# Promoting youth mental health during the COVID-19 pandemic: A longitudinal study

**Maya L. Rosen**[1]*, **Alexandra M. Rodman**[1], **Steven W. Kasparek**[1], **Makeda Mayes**[2], **Malila M. Freeman**[1], **Liliana J. Lengua**[3], **Andrew N. Meltzoff**[2,3], **Katie A. McLaughlin**[1]

**1** Department of Psychology, Harvard University, Cambridge, Massachusetts, United States of America, **2** Institute for Learning & Brain Sciences, University of Washington, Seattle, Washington, United States of America, **3** Department of Psychology, University of Washington, Seattle, Washington, United States of America

\* mayalrosen@fas.harvard.edu

**Data Availability Statement:** All data are available on open science framework https://osf.io/y7cmj/.

## Abstract

The COVID-19 pandemic has introduced novel stressors into the lives of youth. Identifying factors that protect against the onset of psychopathology in the face of these stressors is critical. We examine a wide range of factors that may protect youth from developing psychopathology during the pandemic. We assessed pandemic-related stressors, internalizing and externalizing psychopathology, and potential protective factors by combining two longitudinal samples of children and adolescents (N = 224, 7–10 and 13–15 years) assessed prior to the pandemic, during the stay-at-home orders, and six months later. We evaluated how family behaviors during the stay-at-home orders were related to changes in psychopathology during the pandemic, identified factors that moderate the association of pandemic-related stressors with psychopathology, and determined whether associations varied by age. Internalizing and externalizing psychopathology increased substantially during the pandemic. Higher exposure to pandemic-related stressors was associated with increases in internalizing and externalizing symptoms early in the pandemic and six months later. Having a structured routine, less passive screen time, lower exposure to news media about the pandemic, and to a lesser extent more time in nature and getting adequate sleep were associated with reduced psychopathology. The association between pandemic-related stressors and psychopathology was reduced for youths with limited passive screen time and was absent for children, but not adolescents, with lower news media consumption related to the pandemic. We provide insight into simple, practical steps families can take to promote resilience against mental health problems in youth during the COVID-19 pandemic and protect against psychopathology following pandemic-related stressors.

## Introduction

The COVID-19 pandemic has introduced unprecedented changes in the lives of children and adolescents. These changes brought a sudden loss of structure, routine, and sense of control. Families faced unique stressors ranging from unexpected illness, sudden unemployment and

**Funding:** Funding/Support: This work was supported by the Bezos Family Foundation (to ANM) for collection of data. This work was also supported by the National Institute of Child Health and Human development (F32 HD089514 and K99 HD099203 to MLR) and the National Institute of Mental Health (R01 MH106482 to KAM).

**Competing interests:** The authors have declared that no competing interests exist.

financial stressors, difficulty accessing basic necessities, and increased caretaking responsibilities paired with the shift to remote work, among others [1, 2]. Social distancing guidelines have limited youth's contact with friends, extended family, and teachers, which may increase isolation and loneliness. Schools traditionally provide resources that may buffer youth against the negative consequences of stressors—including supportive social interactions, physical exercise, consistent meals, and a structured routine—that were unavailable to many U.S. youth for a prolonged period of time during the pandemic. These disruptions and pandemic-related stressors are likely to increase risk for depression, anxiety, and behavior problems in youth. Here, we identify factors that may protect against increases in mental health problems during the COVID-19 pandemic in a longitudinal sample assessed both prior to the pandemic and during the stay-at-home order period. We focus on simple and practical strategies that families can take in an effort to promote positive mental health outcomes in children and adolescents during the pandemic.

Exposure to stressors is strongly related to the onset of internalizing and externalizing psychopathology in children and adolescents [3–8]. The powerful association between stress and psychopathology has been replicated in longitudinal studies [7, 9, 10], including following community stressors, such as natural disasters [11, 12] and terrorist attacks [13–15]. Numerous pandemic-related experiences reflect novel stressors for youth and families, including unpredictability and daily routine disruptions [16, 17], unexpected loss of family members, friends, and loved ones [18], chronic exposure to information about threats to well-being and survival in situations that were previously safe [19], and social isolation [20]. Thus, exposure to pandemic-related stressors is likely to be associated with increases in anxiety, depression, and behavior problems in children and adolescents [1, 21, 22]. Indeed, emerging data demonstrates that youth psychopathology has increased during the COVID-19 pandemic [23].

Identifying factors that may promote youth well-being during the pandemic is a critical priority and has clear benefits for parents, pediatricians, and medical professionals. Leading theoretical models of resilience posit that factors that promote resilience exist across multiple levels including the individual, family, school, community, and broader cultural systems [24–26]. Critically, during the early period of the COVID-19 pandemic when schools were closed, stay-at-home orders were in place, and many community resources were shuttered, children were cut off from many common sources of resilience, particularly those occurring at the school and community levels. As such, home and family-level factors may have been of even greater importance than in normal circumstances. Furthermore, given the constraints faced by many families with children, we focus on a set of simple and practical strategies that are easily accessible, inexpensive, and require no specialized resources or services outside the home. We selected factors that have previously been associated with reduced child psychopathology or buffer against mental health problems following exposure to stressors, including: higher levels of physical activity [27–29]; access to nature and the outdoors [30–33]; a consistent daily routine providing structure and predictability [16, 34]; getting a sufficient amount of sleep, which is often disrupted following stressors [35–37]; and lower levels of passive screen time and news media consumption, given that higher use has been associated with elevations in child psychopathology [38], particularly following community-level stressors, like terrorist attacks [39–42]. We also assessed the degree to which youth engaged in adaptive coping strategies during times of distress (e.g., exercising, seeking support from loved ones, or practicing mindfulness or meditation) [43–45]. Finally, providing help for others in need is associated with reduced anxiety and depression [46, 47]. Here, we evaluated whether these nine simple and inexpensive strategies are (a) associated with reduced psychopathology symptoms during the pandemic and (b) buffer against the negative mental health consequences of pandemic-related stressors in children and adolescents.

We examined these questions by combining two longitudinal samples of children and adolescents whose mental health was assessed *prior* to the COVID-19 pandemic in Seattle, Washington. This aspect of this study is critical because one of the strongest predictors of psychopathology during the pandemic is likely to be psychopathology *prior* to the pandemic. By controlling for pre-pandemic psychopathology, we are able to investigate changes in psychopathology that occurred during the pandemic. We then assessed pandemic-related stressors, internalizing and externalizing symptoms, and potential protective factors during six weeks between April and May of 2020—a period when the Seattle area was particularly hard-hit by the pandemic and stay-at-home orders were in place. We also followed up with participants six months later, between late November of 2020 and early January of 2021, to assess mental health. During this second follow-up, schools in the Seattle area were still operating virtually, social distancing guidelines were still in place, and new COVID-19 cases had reached a second peak. We examined whether exposure to pandemic-related stressors were associated with increases in internalizing and externalizing psychopathology, both concurrently and prospectively, controlling for pre-pandemic symptoms. We explored whether the potential protective factors were associated with changes in psychopathology during the pandemic or moderated the association of pandemic-related stressors with changes in psychopathology both during the stay-at-home orders and six months later. Finally, we tested whether these associations varied as a function of age, to determine whether the associations of potentially protective factors with psychopathology were similar for children and adolescents both concurrently and prospectively. Given the unique context of the COVID-19 pandemic, we did not have strong hypotheses about which particular protective factors would be more beneficial to children or adolescents. However, we did hypothesize that adolescents would show a stronger association between pandemic-related stress and psychopathology given previous work that shows that adolescence is a period of particular vulnerability to mental health problems following stressful life events [6, 7, 48–50].

## Methods

### Participants

Participants were recruited from two ongoing longitudinal studies of children and adolescents in the greater Seattle area. A sample of 224 youth aged 7–15 ($M_{age}$ = 12.65, SD = 2.59, range: 7.64–15.24, 47.8% female) and a caregiver completed a battery of questionnaires to assess social behaviors and experiences and pandemic-related stressors. Participants also completed assessments of symptoms of internalizing and externalizing psychopathology. Two participants did not complete these mental health assessments and therefore were excluded from analyses. Six months later, 184 of these youth (82% of the initial pandemic sample) and a caregiver again completed an assessment of internalizing and externalizing symptoms. Ten participants did not complete these mental health assessments and therefore were excluded from analyses at T2. The racial and ethnic background of participants reflected the Seattle area, with 66% of participants identifying as White, 11% as Black, 11% as Asian, 8% as Hispanic or Latino, and 3% as another race or ethnicity.

Children from the first sample were recruited from a study of younger children (*N* = 99) originally recruited between January 2016 and September 2017 [51, 52]. Between March 2018 and November 2018, a subset of the original sample (*N* = 90) participated in a follow-up assessment of mental health. All participants who participated at baseline were contacted for the current study during the period of stay-at-home orders of the pandemic. From this sample, 68 youths (68.9% of the original sample; $M_{age}$ = 8.88, range: 7.64–10.21, 53% female) and a caregiver participated in the first time point of current study (during the stay-at-home orders) and

53 completed the six-month follow-up. Mental health assessments obtained at age 6–8 years were used to control for pre-pandemic psychopathology. Three participants did not complete the most recent assessment, and mental health assessments at age 5–6 were used to control for pre-pandemic psychopathology.

Adolescent participants were drawn from a longitudinal study of children followed from early childhood to adolescence and their mothers [53]. Participants completed the most recent assessment at age 11–12 years ($N$ = 227) between June 2017 and October 2018. These participants were re-contacted for assessment for the current study. From this sample, 154 youths ($M_{age}$ = 14.3, range: 13.12–15.24, 46% female) and their caregiver completed the current study (67.8% of the most recently assessed sample) and 121 completed the six-month follow-up. Mental health assessments at age 11–12 were used to control for pre-pandemic psychopathology.

These two samples came from the same general population (youth in the Seattle area from a wide range of socioeconomic backgrounds). Critically, these two samples did not differ with regards to socioeconomic status, as measured by the income-to-needs ratio, sex distribution (ps > .8), or in exposure to pandemic-related stressors (p = .907).

Participants were excluded from the parent studies based on the following criteria: IQ < 80, active substance dependence, psychosis, presence of pervasive developmental disorders (e.g., autism), and psychotropic medication use. Across both samples, legal guardians provided informed consent and youths provided assent via electronic signature obtained using Qualtrics (Provo, UT). All study procedures were approved by the Institutional Review Board at Harvard University. Youth and their caregivers were each paid $50 for participating in the first wave of the study and $35 for the second wave.

## Procedure

Parents and youth separately completed electronic surveys. Families contacted an experimenter if youth had trouble completing the surveys on their own, and an experimenter then called via phone or video chat and read the questions aloud and recorded their responses (this experimenter was blind to all data from the previous assessments). Data were collected during a six-week period between mid-April, 2020 and May 31st, 2020 (T1), during which schools were closed and stay-at-home orders were in place. A follow-up (T2) was conducted between late November 2020 and early January 2021 in which youth mental health was assessed again.

## Pandemic-related stressors

We developed a set of questions to assess pandemic-related stressors (https://osf.io/drqku/; see S1 File). The assessment included *health*, *financial*, *social*, *school*, *and physical environment stressors* that occurred within the preceding month, based on both caregiver and child report (See Table 1). Given that the COVID-19 pandemic presented a wide range of unique stressors that have not occurred in prior community-wide disruptions, it was necessary to create a novel measure to assess these types of experiences. It is standard practice in the field to do so when novel events occur for which existing stress measures do not adequately capture the full extent of specific types of stressful experiences (*e.g.*, to understand the unique hurricane-related stressors that occurred during Hurricane Katrina or experiences specific to the terrorist attacks on September 11th or the Oklahoma City bombing [12, 41, 54, 55].

We created a composite of pandemic-related stressors using a cumulative risk approach, [56] by determining the presence of each potential stressors (exposed versus not exposed), and creating a risk score reflecting a count of these stressors (18 maximum). Importantly, many previous studies demonstrate the utility and convergent validity of cumulative stress measures

**Table 1. COVID-19 pandemic-related stressors and potential protective factors.**

| Stressor Domain | Description | Number of Items |
|---|---|---|
| Health Stressors | Participant contracted COVID-19; a parent, sibling or another relative contracted COVID-19; a partner or close friend contracted COVID-19; the participant knew someone who died of the virus; had a parent who was an essential worker (*e.g.* healthcare worker, grocery store worker) who was still working during the initial months of the pandemic. | 7 |
| Social | having a difficult relationship with a parent or other member of the household that had gotten worse during the last month; experiencing loneliness a few times per week or more; and experiencing racism, prejudice or discrimination related to the pandemic. | 4 |
| Financial | a parent was laid off or had other significant loss of employment; the family experienced food insecurity, assessed using previously-validated items [81, 82]; the family was evicted or otherwise were forced to leave their home because of financial reasons; the family experienced significant financial loss (*e.g.* due do loss of business, job loss, stock market losses, *etc.*). | 4 |
| School | experiencing difficulty getting schoolwork done at home; the environment where the child does schoolwork is noisy. | 2 |
| Physical Environment | crowding in the home based on the total number of people in the home divided by the approximate square footage reported by the parent [32] | 1 |

| Potential Protective Factor | Description of Measurement |
|---|---|
| Physical Activity | Total minutes of physical activity per week |
| Time in Nature | Days per week they spent time in natural green spaces including parks, canals, nature areas, beaches, countryside, and farmland. |
| Time Outdoors | Days per week participants spend time outside of their home (*e.g.* backyard or neighborhood street) for at least 30 minutes |
| News Consumption | Time spent watching news coverage about the pandemic on a TV, computer, iPad or other electronic device per day. Scored as a binary variable with less than 2 hours per day being scored as 0, and 2 or more hours per day being scored as 1. |
| Passive Screen Time | Hours per day, on average spent watching video on an electronic device, passively scrolling through social media, looking at websites and online news, watching movies and TV. Summed for total passive screen time |

(*Continued*)

**Table 1.** (Continued)

| Stressor Domain | Description | Number of Items |
|---|---|---|
| Sleep Quantity | | Binary measure computed using CDC recommended guidelines for children in this age range (9–12 hours per night for children aged 8–10; 8–9 hours per night for adolescents [83]. |
| Daily Routine | | Participant report on a 4-point Likert scale about the extent to which their days had a fairly consistent routine. |
| Adaptive Coping Strategies | | Binary measure. Participants were given a 1 if they endorsed any of the following ways of dealing with distress related to the coronavirus: talked to family or friends, exercised, meditated, or engaged in self-care activities. |
| Helping in Community | | Binary measure. Participants were given a 1 if they endorsed having participated in any of the following activities: volunteering time at hospitals, donating or preparing food, donating money or supplies, giving shelter to displaced people, praying for others, writing letters or contacting isolated people, cheering on health care workers, or other ways of helping. |

in relation to health outcomes, with a greater number of stressors predicting higher levels of mental and physical health problems [56]. Here, we provide additional evidence for convergent validity by showing that the number of stressors is moderately associated with a measure of perceived stress as measured by the Perceived Stress Scale in this sample (r = 0.399). This value is similar to the correlation between stressful life events and perceived stress observed in the original validity studies used to create the Perceived Stress Scale (r = 0.24-.35) [57].

We also assessed pandemic-related stressors at T2. Importantly we only asked about stressors occurring between T1 and T2. If, for example, a participant had family member who became ill with COVID-19 in April 2020, this would be counted in the pandemic-related stressors at T1, but not at T2. We used pandemic-related stressors at T1 in all analyses (including prospective analyses) but report on pandemic-related stressors at T2 to illustrate the ongoing nature of the pandemic during the second wave of data collection.

## Potential protective factors

We assessed nine potentially protective aspects of youth and family behavior during the prior month: (a) physical activity, (b) time spent in nature, (c) time spent outdoors, (d) screen time, (e) news consumption, (f) sleep quantity, (g) family routines, (h) coping strategies, and (i) helping others (https://osf.io/drqku/, Table 1).

## Internalizing and externalizing psychopathology

Psychopathology was assessed prior to the pandemic by parent and child report on the Youth Self Report (YSR) and Child Behavior Checklist (CBCL) [58, 59]. The CBCL scales are widely used measures of youth emotional and behavioral problems and use normative data to generate age-standardized estimates of internalizing and externalizing psychopathology. We used the highest T-scores from the caregiver or child on the Internalizing and Externalizing symptoms subscales as measures of pre-pandemic symptoms. The children who were 6–8 years old at the pre-pandemic time point did not complete the YSR; only the CBCL was used to compute their pre-pandemic symptoms at that time point. The use of the higher caregiver or child report for psychopathology is an implementation of the standard "or" rule used in combining

caregiver and child reports of psychopathology. In this approach, if either a parent or child endorses a particular symptom it is counted with the assumption that if a symptom is reported, it is likely present. This is a standard approach in the literature on child psychopathology–for example it is how mental disorders are diagnosed in population-based studies of psychopathology in children and adolescents [60, 61].

To assess psychopathology at T1 and T2, parents and youths completed the Strengths and Difficulties Questionnaire, a widely-used assessment of youth mental health [62]. The SDQ has good reliability and validity [63, 64] and correlates strongly with the CBCL/YSR [65]. We chose to use the SDQ to reduce participant burden, as it has substantially fewer items than the CBCL/YSR. We used the highest reported value on the Internalizing and Externalizing symptoms subscales from the caregiver or child.

## Family income

At T1, we asked caregivers to report their total combined family income for the 12 months prior to the onset of the pandemic in 14 bins. The median of the income bins was used except for the lowest and highest bins which were assigned $14,570 and $150,000, respectively. We then calculated the income-to-needs ratio by dividing the family's income by the federal poverty line for a family of that size in 2020, with values less than one indicating income below the poverty line. Nine caregivers did not provide information on family income and were thus excluded from analyses. Median income-to-needs ratio was 4.19 (min = 0.35, max = 8.41).

## Statistical analysis

We used linear regression to investigate the questions of interest. Continuous predictors were standardized using a z-score. Analyses were performed in R using the *lme4* package and standardized coefficients are presented. Continuous age, sex, income-to-needs ratio, and pre-pandemic symptoms measured using the CBCL/YSR prior to the pandemic were included as covariates in all analyses. First, we examined the association of pandemic-related stressors with internalizing and externalizing symptoms, both concurrently and prospectively. Next, we examined the association of potential protective factors with internalizing and externalizing problems, both concurrently and prospectively. Then, we tested whether these factors moderated the association of pandemic-related stressors with psychopathology, both concurrently and prospectively. Finally, we computed interactions of each protective factor with age predicting psychopathology and the interaction of pandemic-related stressors, each potential protective factor, and age predicting psychopathology, both concurrently and prospectively. Simple slopes analysis was used to follow-up on significant interactions using the R *pequod* package. Stratification for simple slope analyses in analyses that used continuous moderators were conducted using a median split. In the case of age analyses, because there was a gap in age between the oldest children (10 years) and the youngest adolescents (13 years), stratifying by sample for these purposes was equivalent to stratifying by a median split. False discovery rate (FDR) correction was applied at the level of hypothesis such that we corrected for comparisons at T1 and T2 (*e.g.*, association between physical activity and internalizing psychopathology at T1 and T2). Listwise deletion was used to handle missing data at T2, excluding participants from analysis who did not complete the second follow-up during the pandemic.

## Results

Prior to the pandemic, 71 participants (31.7% of the sample) were in the subclinical or clinical range for internalizing problems and 39 participants (17.4% of the sample) were in the subclinical or clinical range for externalizing problems. Internalizing and externalizing symptoms

increased substantially during the early phase of the pandemic. Specifically, 127 (56.7%) were in the subclinical or clinical range for internalizing problems and 126 (56.2%) were in the subclinical or clinical range for externalizing problems at the beginning of the pandemic.

See S1 File for the frequency of different domains of stressors at T1 and T2 (S1 Table in S1 File), the distribution of potential protective factors and psychopathology symptoms before and after the pandemic (S2 Table in S1 File), bivariate correlations between all study variables (S3 Table in S1 File) and associations between individual stressors and psychopathology at T1 and T2 (S4 Table in S1 File).

As expected, one of the strongest predictors of psychopathology during the pandemic was pre-pandemic psychopathology (see S3 Table in S1 File). Therefore, it is important to highlight that all analyses controlled for pre-pandemic psychopathology to assess changes in psychopathology specific to the pandemic period.

## Pandemic-related stressors and psychopathology

The number of pandemic-related stressors was strongly associated with increases in both internalizing ($\beta$ = 0.345, $p$ < .001), and externalizing symptoms ($\beta$ = 0.297, $p$ < .001) symptoms during the pandemic, controlling for pre-pandemic symptoms (Fig 1). As expected, pre-pandemic symptoms were also strongly associated psychopathology during the pandemic in this model ($\beta$ = 0.279, $p$ < .001 and $\beta$ = 0.296, $p$ < .001 for internalizing and externalizing psychopathology, respectively).

Similarly, the number of pandemic-related stressors early in the pandemic was positively associated with internalizing ($\beta$ = 0.243, $p$ = .001) and externalizing ($\beta$ = 0.288, $p$ < .001) symptoms later in the pandemic, controlling for pre-pandemic symptoms (Fig 1). Again, pre-pandemic symptoms were strongly associated with internalizing and externalizing problems at T2 ($\beta$ = 0.260, $p$ = .001 and $\beta$ = 0.278, $p$ < .001, respectively).

The association of pandemic-related stressors with internalizing symptoms varied by age ($\beta$ = 0. 0.602, $p$ = .043), such that the association was stronger among adolescents (simple slope: b = 0.437, $p$ < .001) than children (simple slope: b = 0.220, $p$ = .004) concurrently. There were interactions between age and pandemic-related stressors in predicting externalizing symptoms concurrently or prospectively.

## Potential protective factors

Associations of potential protective factors with concurrent psychopathology and interactions with stress and age are summarized in Table 2. Associations of potential protective factors with prospective psychopathology and interactions with stress and age are summarized in Table 3.

**Physical activity.** Physical activity was unrelated to psychopathology concurrently or prospectively.

**Time spent in nature and outdoors.** Greater time spent in nature was marginally associated with lower internalizing problems both concurrently and prospectively (Fig 2A and 2B), controlling for pre-pandemic symptoms. Time spent outdoors was unrelated to psychopathology. Age did not moderate any of these associations.

**News consumption and passive screen time.** Early in the pandemic, youths who spent less time on digital devices each day had lower externalizing symptoms (Fig 2C and 2D), controlling for pre-pandemic symptoms. Consuming <2 hours of news per day was also associated with reduced externalizing symptoms early in the pandemic (Fig 2G).

The longitudinal association between screen time and internalizing symptoms varied by age (S1 Fig in S1 File), such that children showed a positive association between screen time

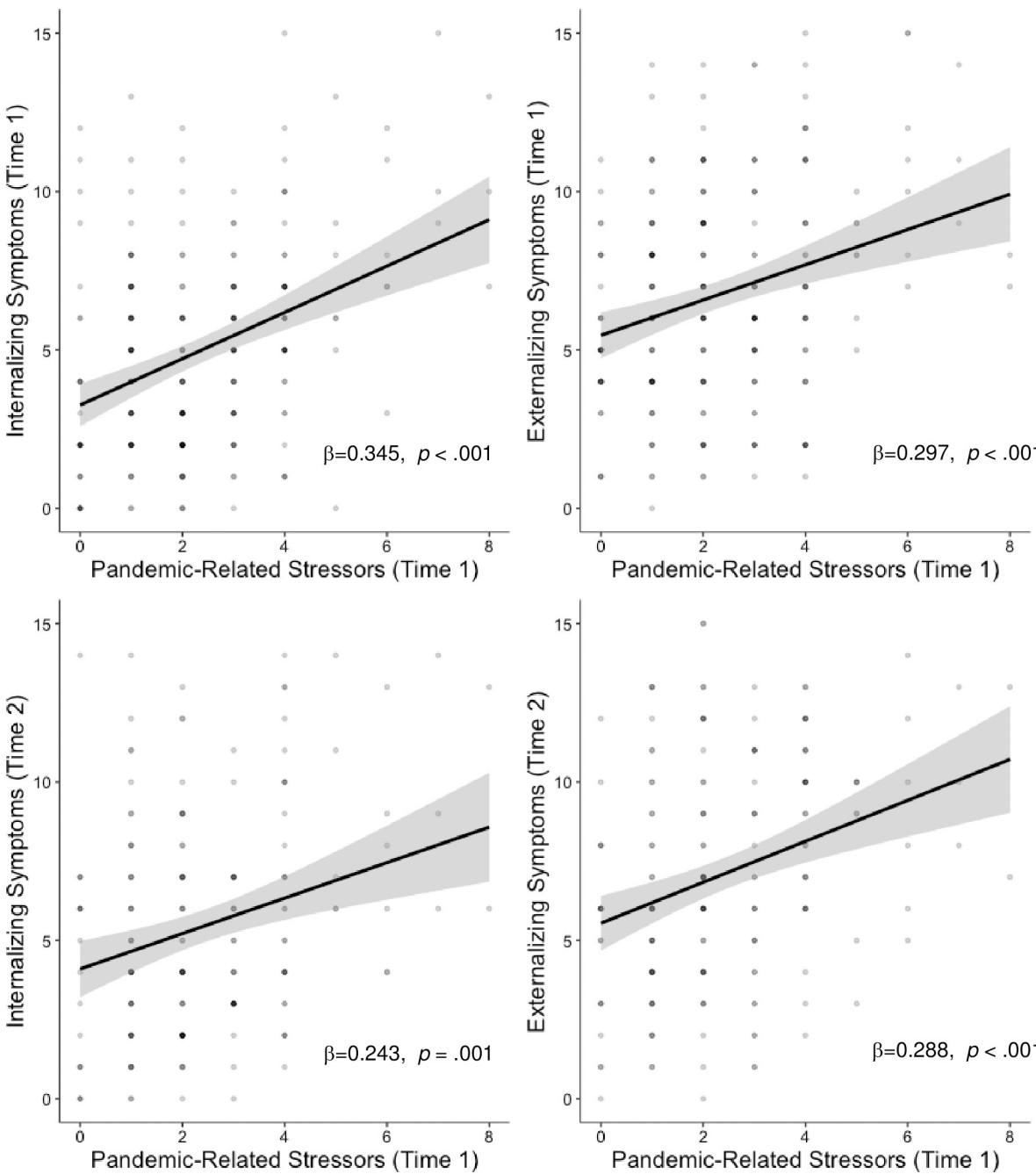

**Fig 1. Main effects of pandemic-related stressors and psychopathology.** All analyses control for age, sex, income-to-needs and pre-pandemic psychopathology symptoms.

and internalizing psychopathology six months later (b = 0.572, $p$ = .008), but adolescents did not (b = -0.074, $p$ = .512).

Age moderated the association between news consumption and internalizing psychopathology prospectively. Specifically, while children showed a positive association between news consumption and internalizing psychopathology at T2 (b = 0.438, $p$ = 0.015), adolescents showed a negative association between news consumption and internalizing psychopathology at T2 (b = -0.299, $p$ = .015).

**Table 2. Associations of potential protective factors with psychopathology and interactions with stress and age at T1.** Significant associations are presented in BOLD and marginal associations are presented in italics.

| Protective Factors | | Internalizing | | Externalizing | | Age Internalizing Interaction | | Age Externalizing Interaction | |
|---|---|---|---|---|---|---|---|---|---|
| | | β | p | β | p | β | p | β | p |
| Physical Activity | Main effect | -0.120 | 0.132 | -0.016 | 0.900 | 0.075 | 0.826 | 0.391 | 0.243 |
| | Stress Interaction | -0.153 | 0.191 | -0.146 | 0.448 | 0.151 | 0.790 | 0.609 | 0.588 |
| Time in Nature | Main effect | *-0.124* | *0.074* | 0.029 | 0.777 | 0.045 | 0.885 | 0.139 | 0.658 |
| | Stress Interaction | -0.067 | 0.602 | 0.013 | 0.913 | -0.580 | 0.271 | 0.218 | 0.682 |
| Time Outdoors | Main effect | 0.000 | 0.999 | 0.018 | 0.779 | -0.455 | 0.144 | -0.215 | 0.484 |
| | Stress Interaction | -0.088 | 0.846 | -0.238 | 0.112 | -0.146 | 0.793 | 0.613 | 0.381 |
| Passive Screen Time | Main effect | 0.059 | 0.431 | **0.272** | **0.0004** | *-1.084* | *0.074* | *-0.979* | *0.087* |
| | Stress Interaction | **0.561** | **0.002** | **0.329** | **0.050** | -1.399 | 0.368 | 0.729 | 0.531 |
| News Consumption | Main effect | 0.093 | 0.374 | **0.193** | **0.010** | *-0.741* | *0.083* | -0.312 | 0.453 |
| | Stress Interaction | *0.273* | *0.074* | 0.197 | 0.136 | **-1.474** | **0.028** | 0.389 | 0.771 |
| Sleep Quantity | Main effect | -0.018 | 0.995 | -0.061 | 0.370 | 0.674 | 0.130 | 0.551 | 0.126 |
| | Stress Interaction | -0.171 | 0.326 | 0.094 | 0.762 | *-1.728* | *0.064* | -0.623 | 0.451 |
| Daily Routine | Main effect | -0.022 | 0.736 | *-0.122* | *0.058* | -0.062 | 0.854 | -0.011 | 0.974 |
| | Stress Interaction | -0.197 | 0.211 | -0.131 | 0.535 | 0.206 | 0.766 | 0.214 | 0.763 |
| Adaptive Coping | Main effect | 0.061 | 0.688 | 0.124 | 0.102 | -0.436 | 0.377 | -0.040 | 0.906 |
| | Stress Interaction | 0.177 | 0.276 | -0.083 | 0.488 | -0.587 | 0.630 | *1.225* | *0.078* |
| Helping | Main effect | 0.002 | 0.978 | 0.012 | 0.848 | 0.401 | 0.231 | 0.186 | 0.575 |
| | Stress Interaction | -0.059 | 0.623 | -0.081 | 0.968 | -0.348 | 0.571 | 0.674 | 0.281 |

**Table 3. Associations of potential protective factors with psychopathology and interactions with stress and age at T2.** Significant associations are presented in BOLD and marginal associations are presented in italics.

| Protective Factors | | Internalizing | | Externalizing | | Age Internalizing Interaction | | Age Externalizing Interaction | |
|---|---|---|---|---|---|---|---|---|---|
| | | β | p | β | p | β | p | β | p |
| Physical Activity | Main effect | -0.049 | 0.515 | 0.009 | 0.900 | -0.126 | 0.826 | 0.534 | 0.243 |
| | Stress Interaction | -0.241 | 0.179 | 0.028 | 0.816 | 0.712 | 0.686 | -0.182 | 0.802 |
| Time in Nature | Main effect | *-0.136* | *0.074* | -0.021 | 0.777 | -0.264 | 0.885 | 0.371 | 0.516 |
| | Stress Interaction | 0.077 | 0.602 | 0.069 | 0.913 | -0.706 | 0.271 | -0.869 | 0.300 |
| Time Outdoors | Main effect | -0.048 | 0.999 | 0.066 | 0.750 | -0.566 | 0.144 | -0.351 | 0.484 |
| | Stress Interaction | 0.029 | 0.846 | -0.163 | 0.254 | 0.543 | 0.793 | 0.568 | 0.381 |
| Passive Screen Time | Main effect | 0.097 | 0.431 | *0.157* | *0.076* | **-1.953** | **0.030** | *-1.264* | *0.087* |
| | Stress Interaction | **0.401** | **0.049** | **0.606** | **0.003** | -1.243 | 0.368 | 1.158 | 0.531 |
| News Consumption | Main effect | -0.040 | 0.627 | 0.114 | 0.152 | **-1.743** | **0.004** | -0.932 | 0.170 |
| | Stress Interaction | 0.034 | 0.829 | 0.223 | 0.136 | **-2.199** | **0.018** | 0.238 | 0.771 |
| Sleep Quantity | Main effect | 0.000 | 0.995 | *-0.158* | *0.080* | 0.299 | 0.479 | *0.682* | *0.126* |
| | Stress Interaction | 0.045 | 0.761 | 0.043 | 0.762 | *-1.685* | *0.089* | -1.562 | 0.184 |
| Daily Routine | Main effect | 0.034 | 0.736 | **-0.164** | **0.049** | -0.129 | 0.854 | 0.589 | 0.238 |
| | Stress Interaction | -0.191 | 0.221 | 0.092 | 0.535 | -0.648 | 0.766 | -0.865 | 0.666 |
| Adaptive Coping | Main effect | -0.015 | 0.845 | 0.103 | 0.156 | -0.353 | 0.377 | -0.152 | 0.906 |
| | Stress Interaction | 0.099 | 0.506 | 0.153 | 0.488 | -0.149 | 0.849 | 0.218 | 0.767 |
| Helping | Main effect | 0.036 | 0.978 | 0.026 | 0.848 | *0.835* | *0.064* | -0.237 | 0.575 |
| | Stress Interaction | -0.088 | 0.623 | -0.005 | 0.968 | 1.296 | 0.186 | 1.054 | 0.281 |

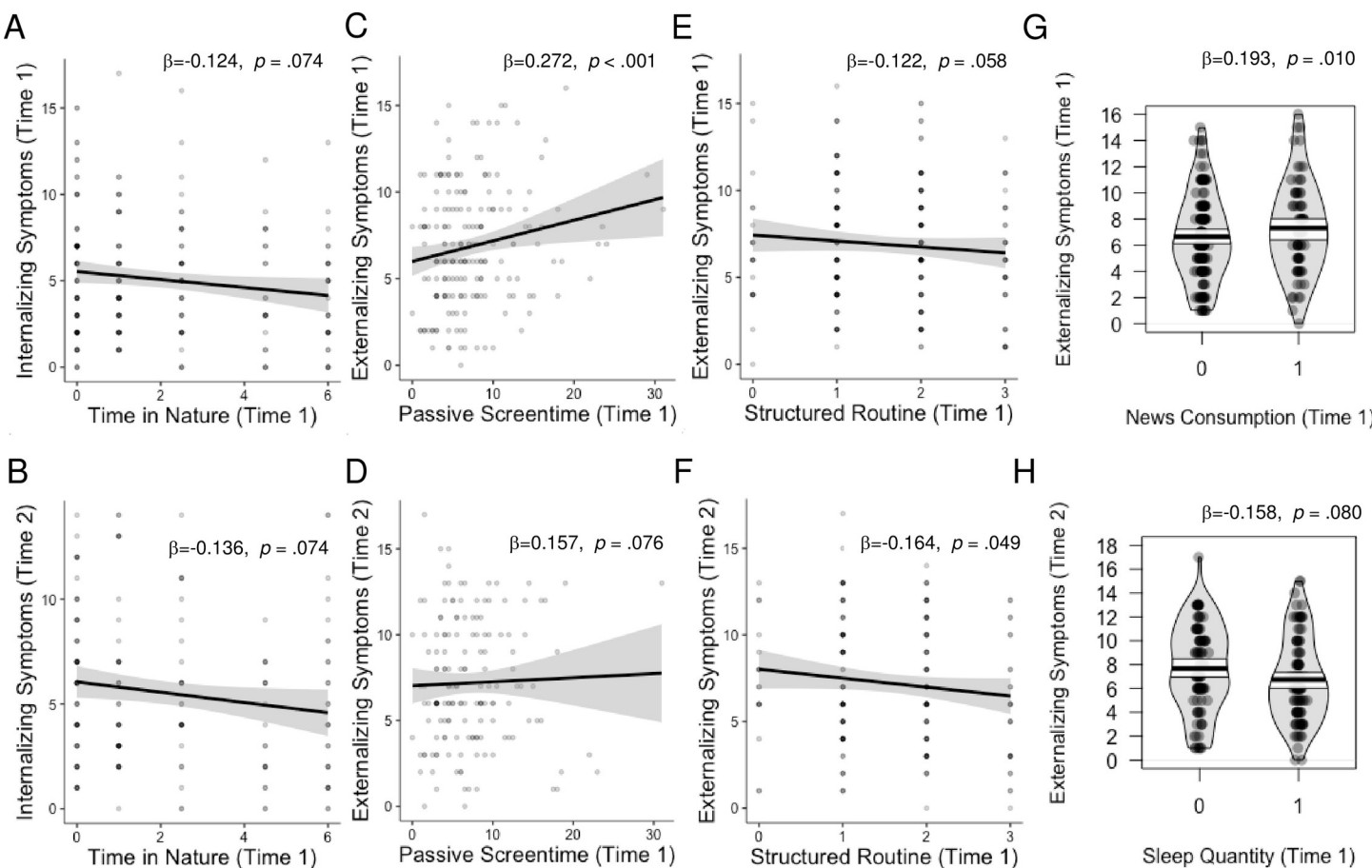

**Fig 2. Main effects of protective factors on psychopathology.** All analyses controlled for age, sex, income-to-needs, and pre-pandemic psychopathology symptoms.

Screen time moderated the association of pandemic-related stressors with internalizing and externalizing psychopathology concurrently and prospectively (Fig 3). Specifically, youths who spent more time on screens showed a strong positive association of pandemic-related stressors with concurrent (b = 0.513, $p < .001$) and prospective (b = .335, $p < .001$) internalizing symptoms as well as both concurrent (b = 0.285, $p < .001$) and prospective (b = .383, $p < .001$) externalizing problems that was absent for youths who spent less time on screens at both time points (b = 0.020–0.061, $p = .445$-.935).

A three-way interaction was observed between news consumption, age, and pandemic-related stressors in predicting internalizing symptoms both concurrently and prospectively (Fig 4). Pandemic-related stressors were unrelated to internalizing problems concurrently (b = -.087, $p = .502$) or prospectively (b = -0.036, $p = .808$) among children who consumed <2 hours of news media per day, but were strongly associated with internalizing psychopathology both concurrently (b = 0.39 2, $p < .001$) and prospectively (b = 0.328, $p = .026$) among children with >2 hours daily news consumption. Among adolescents, pandemic-related stressors were strongly associated with internalizing problems concurrently (b = 0.409–0.452, $p < .001$), regardless of news consumption. Adolescents who consumed low levels of news during the stay-at-home orders showed a positive association between pandemic-related stressors and internalizing psychopathology six months later (b = 0.509, $p = .002$), while adolescents who consumed more news did not (b = 0.113, $p = .346$).

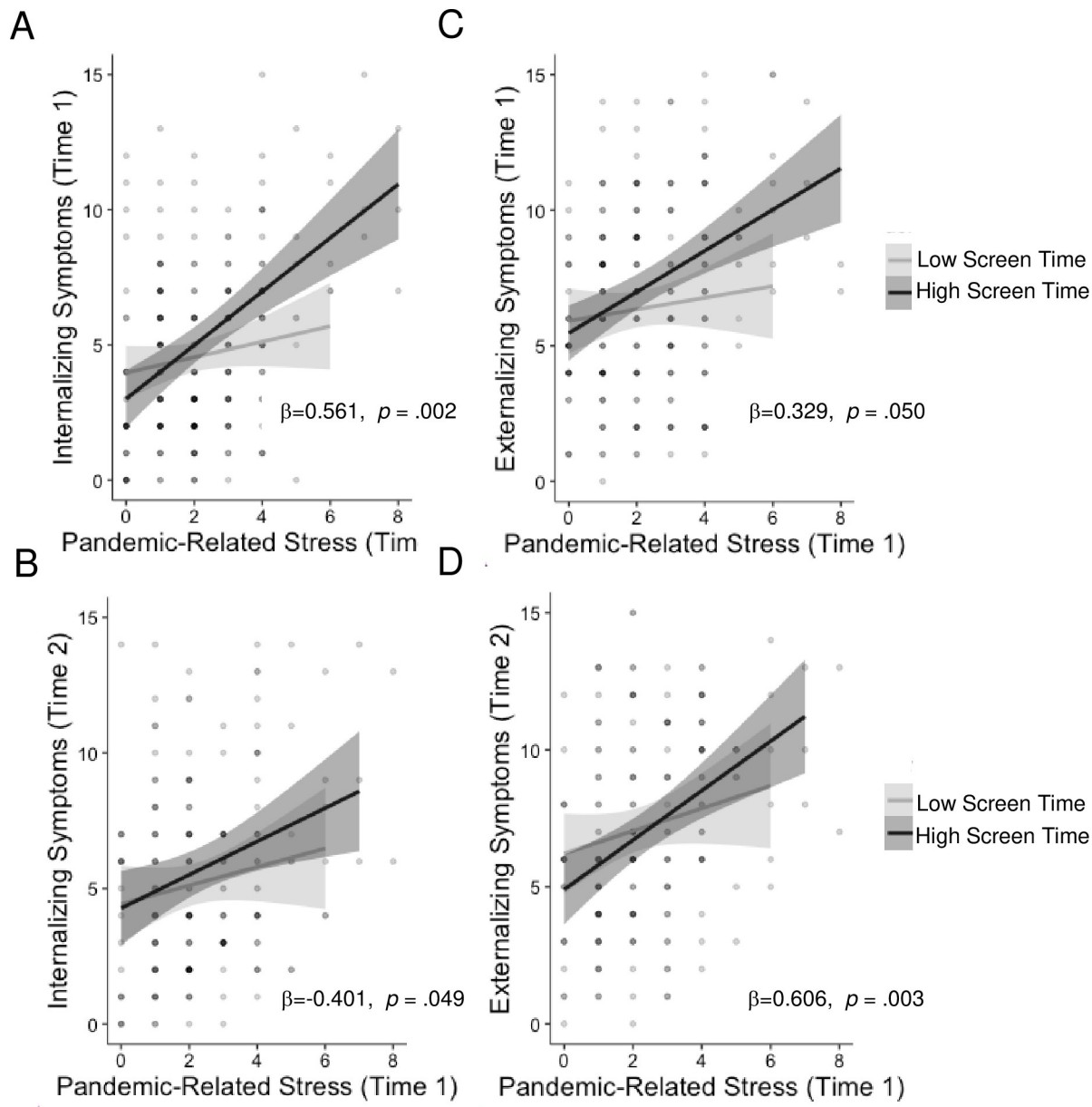

**Fig 3. Passive screen time x stress interaction.** Low screen time use buffers against pandemic-related increases in internalizing and externalizing psychopathology. All analyses control for age, sex, income-to-needs ratio, and pre-pandemic psychopathology symptoms.

**Sleep quantity.** Getting the recommended number of hours of sleep was unrelated to psychopathology concurrently. However, getting the recommended amount of sleep during the stay-at-home orders was marginally associated with lower levels of externalizing psychopathology six months later, controlling for pre-pandemic symptoms (Fig 2H). These associations did not vary by age.

**Routine.** Youths with a more structured daily routine had lower externalizing (Fig 2E and 2F) six months later. No associations of a structured routine were found with internalizing symptoms, and no interactions with age or stress emerged.

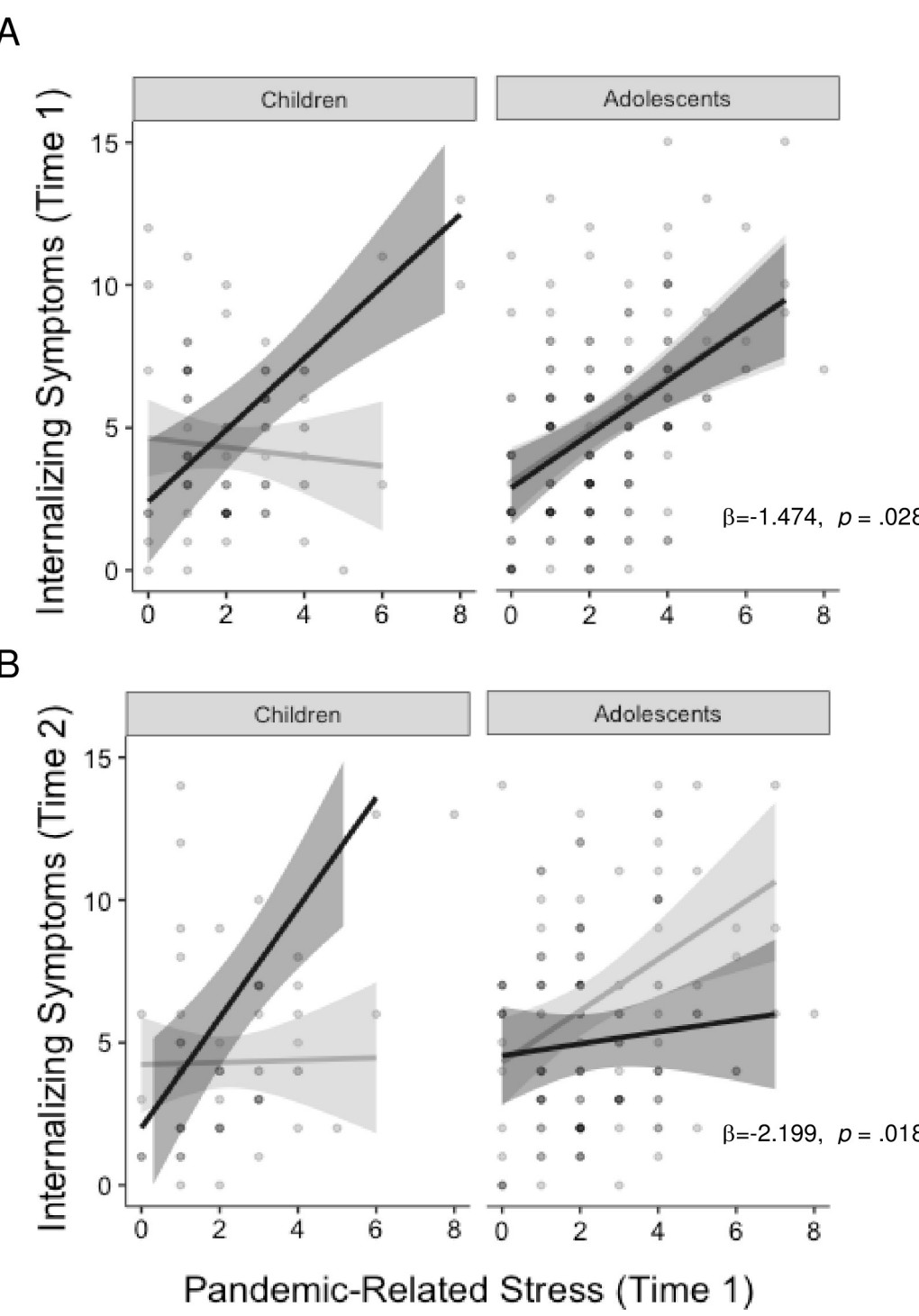

**Fig 4. Age x stress x news interaction.** Low news consumption buffers children, but not adolescents, against pandemic-related increases in internalizing psychopathology concurrently (A) and prospectively (B). All analyses control for age, sex, income-to-needs ratio, and pre-pandemic psychopathology symptoms.

**Coping strategies.** There was no significant association between engaging in adaptive coping strategies with psychopathology concurrently or prospectively.

**Helping others.**   Helping in one's community was unrelated to psychopathology concurrently or prospectively.

## Discussion

The present study identifies simple and practical behaviors that are associated with well-being among children and adolescents during the COVID-19 pandemic. Critically, this study involved a longitudinal sample of children and adolescents for which mental health had been assessed prior to the pandemic, during the stay-at-home orders, and six months later allowing us to investigate psychopathology during the pandemic while controlling for pre-pandemic symptoms. As expected, we found that youths who experienced greater pandemic-related stressors had higher levels of internalizing and externalizing psychopathology. Importantly, greater pandemic-related stressors during the stay-at-home orders were also prospectively associated with higher levels of both internalizing and externalizing psychopathology six months later. Critically, we identified several factors—including a structured daily routine, low passive screen time use, low news media consumption about the pandemic, and to a lesser extent spending more time spent in nature and getting the recommended amount of sleep— that are associated with better mental health outcomes in youth during the pandemic. We additionally demonstrate that the strong association between pandemic-related stressors and psychopathology is absent among children with lower amounts of screen time and news media consumption.

Youth who had a structured and predictable daily routine were less likely to experience increases in externalizing problems during the pandemic than youth with less structured routines. A sudden loss of routine has occurred for many families during the pandemic related to school closures, changes in parental work arrangements, and loss of access to activities outside the home for youth and adolescents. These disruptions in daily routine are associated with increased risk for behavior problems in youth during the pandemic, consistent with prior work suggesting that lack of predictability is strongly linked to youth psychopathology [16, 34, 66, 67]. Moreover, a recent paper during the pandemic showed that preschoolers in families that maintained a structured routine during the pandemic showed lower rates of depression and externalizing problems, over and above the effect of food insecurity, socioeconomic status, dual-parent status, maternal depression, and stress [68]. Our current findings extend this work by demonstrating that a structured routine may also be important for older children and adolescents. Although maintaining routine and structure is challenging as school closures continue and many aspects of daily life remain unpredictable, creating a structured daily routine for children and adolescents may promote better mental health during the pandemic.

Greater passive screen time use was associated with higher levels of externalizing psychopathology early in the pandemic, and greater passive screen time use was associated with higher internalizing psychopathology later in the pandemic for children but not adolescents. Additionally, the strong association of pandemic-related stressors with internalizing and externalizing psychopathology both concurrently and prospectively was reduced in children and adolescents with low passive screen time use. Previous studies have argued that the increases in screen time use over the last decade may be responsible for rising levels of anxiety and depression among children and adolescents [38]. However, others have suggested that greater screen time use may not have negative impacts [69, 70] and that psychopathology and digital device use have a reciprocal association with one another [71]. During the pandemic, youths were encouraged to use digital devices more than ever for school and social connection, which are likely to be beneficial for their development. Here, we measured *passive use* of digital devices, including watching videos on an electronic device, passively scrolling through social

media, looking at websites and online news, and watching movies and TV—excluding more active uses of digital devices for schooling and social communication. Greater research is needed to determine whether the amount of passive screen time itself has negative effects on youth mental health or whether this association simply reflects that greater time on digital devices takes time away from other important behaviors such as exercise, sleep, or connecting with friends or family. Indeed, in the present study, screen time was inversely related with sleep quantity (S3 Table in S1 File). Therefore, one reason that youths with lower screen time use may be buffered against pandemic-related increases in psychopathology is because they are engaging in other behaviors that promote well-being such as getting sufficient sleep, among others. Together, these findings suggest some potential benefits associated with limiting *passive* screen time among youth during the pandemic.

Our findings also suggest that limiting news consumption about the pandemic may be beneficial, particularly for younger children. Greater news media consumption about the pandemic was associated with higher levels of externalizing problem early in the pandemic. Moreover, the strong association between pandemic-related stressors and internalizing psychopathology was absent in children who consumed lower levels of news media, although pandemic-related stressors were positively associated with internalizing symptoms in adolescents regardless of news consumption concurrently. This finding is broadly consistent with previous studies observing strong associations between media exposure about community-level stressors, including terrorist attacks and natural disasters, and higher rates of psychopathology in children and adolescents [41, 42, 72–74]. Interestingly, the same pattern persisted for children six months into the pandemic, while for adolescents who consumed more news during the stay-at-home orders showed a weaker association between stress and internalizing psychopathology six months later than those who consumed less news. Therefore, it is possible that for adolescents, having more knowledge about the pandemic early on may have been beneficial over time. Together these findings suggest that limiting certain types of news media exposure may protect against pandemic-related increases in internalizing problems, especially among young children. Importantly, this does not imply that parents should refrain from discussing the pandemic or hide the realities from their children. In fact, previous studies have found that honest conversations between parents and children provide an important protection against the development of psychopathology in the wake of natural disasters [75]. Therefore, we suggest limiting sensational news media consumption, in favor of talking to children about what is happening, listening to their concerns, and answering their questions in an age-appropriate manner.

Additionally, we found weaker and only marginally significant associations between time spent in nature and getting the recommended amount of sleep with youth psychopathology during the pandemic. We briefly discuss these findings here, as they highlight additional strategies that could be beneficial to families when considering how to support the mental health of their children during the pandemic. Greater time spent in nature was marginally associated with lower increases in internalizing symptoms relative to pre-pandemic symptoms both concurrently and prospectively. These findings are broadly consistent with prior evidence that spending at least two hours in nature per week is associated with greater well-being in adults [31] and better mental health in children [76]. Additionally, the association of stressors with well-being is reduced among children with greater access to nature [77]. Encouraging youths to spend time in nature may also be beneficial for mental health during the pandemic. In addition, children and adolescents who got the recommended amount of sleep at the beginning of the pandemic showed marginally lower levels of externalizing psychopathology six months later. These findings highlight the importance of encouraging youths to get an adequate amount sleep. Given the negative association between screen time and sleep duration both

here and in prior work [78], reducing access to digital devices prior to bedtime may be one simple strategy parents can use to make it easier for their children to get an adequate amount of sleep.

## Limitations

The present study has several limitations which should be acknowledged. First, we relied on self-report measures of behavior, which can be inaccurate due to recall bias. Future studies may benefit from using actigraphy to assess physical activity and sleep, geolocation to measure time spent in nature and outdoors, and direct reports of screen time use and news media consumption from digital devices for more accurate measures of potential protective factors. Second, while the longitudinal nature of the present study is a strength, it only included two snapshots of youth behavior and mental health during the pandemic. It will be important to continue to follow youths throughout the pandemic to determine factors that promote long-term risk and resilience. Third, we used a different measure of psychopathology prior to the pandemic (CBCL/YSR) than after the onset of the pandemic (SDQ). While it would have been ideal to have the same measure at all time points, the CBCL/YSR is much longer than the SDQ and we were focused on minimizing participant burden during a period of time when families were facing numerous stressors and loss of access to typical childcare options. Thus, we chose to use a shorter questionnaire that is strongly correlated with the CBCL/YSR [62, 65, 79, 80]. Relatedly, we asked questions about potential protective factors in our COVID Experiences Survey, rather than using longer validated scales for each of the factors (*e.g.* Pittsburgh Sleep Quality Index, Physical Activity Questionnaire for Children, Media Parenting Practices, Family Routines Inventory, German Coping Questionnaire for 'Children and Adolescents, etc.). This choice was made to maximize the information gained about each family, while minimizing participant burden and thus maximizing our sample size. Fourth, we combined data from two separate samples of children (aged 7–10 and 13–15 at T1). Both samples were recruited using similar methods from the same target population, and we had identical measures of pre-pandemic psychopathology on both samples. Moreover, the samples did not differ in demographics, SES, or exposure to pandemic-related stressors. However, using two samples with a gap in age limited our ability to understand age effects across the entire spectrum of childhood and adolescence. Fifth, we demonstrate the predictive validity of the pandemic-related stress measure via moderate associations with psychopathology at both waves as well as a measure of perceived stress. However, this cumulative risk approach is limited in that it weights stressors equally that could have variable impacts. Future work should investigate whether specific stressors have been more strongly linked to changes in mental health during the pandemic (see S4 Table in S1 File for associations of specific stressors and psychopathology at T1 and T2). Finally, the present study is correlational and we are therefore limited in our ability to make causal inferences about the factors that promote well-being during the pandemic. However, given the extensive literature about the links between these factors and youth mental health, there is little reason to expect downsides to encouraging families to engage in these types of protective behaviors with their children and adolescents during the pandemic.

## Conclusions and practical implications

We identify practical and easily accessible strategies that may promote greater well-being for children and adolescents during the COVID-19 pandemic. Based on these findings, we suggest that parents encourage youth to develop a structured daily routine, limit passive screen time use, limit exposure to news media—particularly for young children, and to a lesser extent spend more time in nature, and encourage youth to get the recommended amount of sleep.

## Supporting information

**S1 File. Please see the S1 File for frequencies of exposures to pandemic-related stressors by domain (S1 Table), distribution of potential protective factors and psychopathology symptoms (S2 Table), bivariate correlation table of all study variables (S3 Table), associations between individual stressors and psychopathology symptoms (S4 Table), age by screen time interaction predicting internalizing symptoms (S1 Fig), and the full COVID experiences surveys (caregiver and child).**
(DOCX)

## Acknowledgments

The authors would also like to acknowledge Reshma Sreekala for help with data collection and Frances Li for help with compiling surveys.

## Author Contributions

**Conceptualization:** Maya L. Rosen, Alexandra M. Rodman, Steven W. Kasparek, Andrew N. Meltzoff, Katie A. McLaughlin.

**Data curation:** Steven W. Kasparek.

**Formal analysis:** Maya L. Rosen.

**Funding acquisition:** Andrew N. Meltzoff, Katie A. McLaughlin.

**Methodology:** Maya L. Rosen, Alexandra M. Rodman, Steven W. Kasparek, Malila M. Freeman, Andrew N. Meltzoff, Katie A. McLaughlin.

**Project administration:** Maya L. Rosen, Alexandra M. Rodman, Steven W. Kasparek, Makeda Mayes.

**Supervision:** Maya L. Rosen, Alexandra M. Rodman, Steven W. Kasparek.

**Writing – original draft:** Maya L. Rosen.

**Writing – review & editing:** Maya L. Rosen, Alexandra M. Rodman, Makeda Mayes, Malila M. Freeman, Liliana J. Lengua, Andrew N. Meltzoff, Katie A. McLaughlin.

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
