## [Decision Letter · Decision Letter 0]

19 May 2021

PONE-D-21-09922

Promoting youth mental health during COVID-19: A Longitudinal Study spanning pre- and post-pandemic

PLOS ONE

Dear Dr. Rosen,

Thank you for submitting your manuscript to PLOS ONE. After careful consideration, we feel that it has merit but does not fully meet PLOS ONE’s publication criteria as it currently stands. Therefore, we invite you to submit a revised version of the manuscript that addresses the points raised during the review process.

Unfortunately, no one of the suggested reviewers agreed to provide comments. Therefore, I invited new reviewers; some of them declined the invitation to review, others have not responded. All this caused the unwanted delay in reviewing your paper. The number of reviews required for your manuscript has been received on May 16^th^. Comments from the two reviewers appear below. After reading the paper myself, I would suggest you to consider the presentation and discussion of the findings for a three-way age x coping x stress interaction .

On page 15, the 1^st^ paragraph states that adolescents, in contrast to young children, showed the opposite pattern of associations between pandemic-related stress and externalizing problems. However, the description of the findings for children and adolescents seems very similar. The Discussion, lines 441-443 states: “children who engaged in more coping strategies showed a stronger association between stress and externalizing symptoms concurrently”. This seems opposite to the findings reported on lines 335-336. I would suggest to present and discuss this finding more clearly. A figure presenting this three-way interaction would be very helpful to the reader.

Discussion, lines 359-360: “the strong association between pandemic-related stressors and psychopathology is absent” – please reformulate to make it clearer.

We look forward to receiving your revised manuscript.

Kind regards,

Helena R. Slobodskaya, M.D., Ph.D., D.Sc.

Academic Editor

PLOS ONE

Journal Requirements:

Reviewers' comments:

Reviewer's Responses to Questions

**Comments to the Author**

1. Is the manuscript technically sound, and do the data support the conclusions?

Reviewer #1: Partly

Reviewer #2: Partly

2. Has the statistical analysis been performed appropriately and rigorously? 

Reviewer #1: Yes

Reviewer #2: No

3. Have the authors made all data underlying the findings in their manuscript fully available?

Reviewer #1: Yes

Reviewer #2: Yes

4. Is the manuscript presented in an intelligible fashion and written in standard English?

Reviewer #1: Yes

Reviewer #2: Yes

5. Review Comments to the Author

Reviewer #1: The present manuscript investigated longitudinal effects of stressors related to the COVID-19 pandemic on youth’s (7-15 years) psychopathological development as well as possible protective factors. Stressors were positively related to internalizing and externalizing symptoms half a year later. Exercise, a structured routing, and less screen time were related to less psychopathological symptoms. Authors discuss practical steps promoting the positive adjustment of families and youth to the stressful environment of the COVID-19 pandemic.

Overall evaluation

The manuscript addresses an important practical question on the psychological factors benefitting youth during the exposure to COVID-19 related stressors. However, the manuscript suffers from limitations mainly concerning theoretical background, hypotheses, methodology, and the results.

Major Concerns

Literature Review: The review presents a host of findings linking stress factors to youth’s psychopathological symptoms. However, the deduction of hypotheses seems largely theoretically unmotivated. Given the uniquely stressful environment caused by the COVID-19 pandemic, developmental theories should be included to predict expected effects and age differences. For example, while the inclusion of potentially protective factors understandably relies on practicability, it would be important to get more insights into their relation to developmental theories.

In addition, hypotheses should be more clearly specified with respect to the expected effects, especially concerning potential age differences between children and youth.

Finally, it seems questionable to call the follow-up measurement a post-pandemic assessment, given that many stressors may persist into this timeframe. Authors should give a more detailed overview over the respective pandemic-related restrictions and social-distancing measures so that one can get an idea of the exposure to stressors during the second measurement point. Otherwise the findings might be explained by youth’s increasing adaptation to the ongoing pandemic as much as by a change in pandemic-related stressors.

Method: The creation of the scales assessing potential protective factors and pandemic-related stressors should be described in more detail (e.g., reliability). Given that protective factors seem less pandemic-specific than the stressors, why did authors not rely on an established scale?

What was the rationale behind selecting the highest values of caregiver or child on the CBCL and the SDQ? This seems at odds with the way theses scales are usually used and introduces the problem that some values are self-reported (if youth values were chosen) while other values are reports from caregivers.

In addition, why was the measure of psychopathology not kept constant from the pre-pandemic assessment? The comparison with pre-pandemic levels of psychopathology would be more informative if both measurement points had used the same measure.

Results: Running separate analyses for each of the protective factors seems questionable and also gives rise to the problem of inflating the alpha-error probability. Perhaps one could combine protective factors based on their correlational structure to obtain less factors overall. This seems also more feasible with the composite score of pandemic-related stressors that authors describe in the methods.

In addition, it would be interesting to see how levels of psychopathology changed from pre-pandemic to T1 to get insights into a possible increase in psychopathological symptoms after the onset of the pandemic.

Minor

p. 6 line 135: typo - drawn

p. 6 lines 130 and 139: “their” instead of “a” careviger

Reviewer #2: Thank you for the opportunity to review this interesting, timely, and well-written manuscript. In this paper, the authors provide a thoughtful examination of which protective (and easily modifiable) factors in the child’s ecology may mitigate internalizing and externalizing symptom distress during the COVID-19 pandemic. The introduction is well formulated and provides a good overview of the literature. There are many strengths to this paper, including its longitudinal design with pre-post measurements and two COVID-19 time points, practical approach to identifying modifiable protective factors, and multi-informant data. However, my enthusiasm was tempered by the amalgamation of two samples varying by developmental period and some of the analytic choices. Below I provide some suggestions to increase clarity, transparency, and impact.

Certainly, a major strength of the paper is the availability of pre-pandemic internalizing and externalizing symptoms and controlling for these in analyses. While the authors provide strong justification in the introduction for why stress should increase during COVID-19 and for their moderator variables, I do think it should be noted in the intro that the strongest predictor of pandemic internalizing/externalizing problems is likely to be pre-pandemic mental health difficulties. This is implied but not explicitly stated in the introduction and is evident when exploring the correlation matrix where associations between baseline (pre-pandemic) and COVID-19 internalizing and externalizing are strong (and much larger than any of the other variables included). This finding should also be noted at the outset of the results and/or in the discussion because it contextualizes the findings.

The authors aptly note several factors that may be associated with child stress and increased behavior problems during COVID-19. These could all independently predict child outcomes. What is the justification for combining these into a cumulative risk composite? From a developmental psychopathology perspective, I understand, but the justification is needed nonetheless, especially for those less familiar with cumulative risk.

It is not until the methods that one discovered the sample is in fact two separate longitudinal samples. Greater justification is needed to combine two different samples, especially with such different age ranges and developmental periods. The authors should be more transparent in the abstract and the introduction about this fact, and also should justify why these samples should be combined (other than to increase sample size).

Methods

1. “174 of these youth” add (77.7% of original sample) so that attrition is clearly stated.

2. Do these two samples differ on any demographic or levels of the stress or protective factors? These should be tested and noted and sign differences should be included as covariates.

3. Going along with point #2, did all children provide self-reports of behavior problems? I am assuming that the 6-8 year olds did not pre-pandemic. This needs to be made clear as the multi-informant data is a strength, but it’s not clear how extensive this multi-informant data stretches.

4. Why the switch from the CBCL to the SDQ?

5. Using the highest t-score for either rater – is there support for this in the literature? Why not average informants?

6. What are the alphas for the behavioral problem measures?

7. With close to 25% of the sample lost at the COVID follow up, how was missing data handled?

8. Adding child sex and pre-pandemic mental health is a strength of this paper. But why not add child ethnicity and family income, for of which predict the outcomes in this paper. I also think age should be added as a covariate (versus stratifying by age).

9. Given the number of analyses run, the authors should adjust the p-value for multiple contrasts.

Results

1. Are patterns of associations similar across the two samples?

2. “The number of pandemic-related stressors was strongly associated with increases in both internalizing (�=0.322 [0.211, 0.432], and externalizing symptoms (�=0.225 [0.136, 0.314], 208 p<.001), symptoms during the pandemic, controlling for pre-pandemic symptoms”. Can you add the estimates here for the pre-pandemic symptoms so that the magnitude of effects can be directly evaluated (by the reader). As far as I can tell, this data is not in any of the tables/figures. Please report the pandemic-related stressors to psychopathology analyses in a table and include covariates to ensure the findings hold with these variables included. This is especially important given the sample differences.

3. Are the stratification analyses by age simply a stratification by the two samples? If so, they should be presented as such for more clarity and age as a continuous measure should be used a co-variate in analyses.

Discussion

1. “mental health had been carefully assessed prior to the pandemic” remove carefully here as that implies diagnostic interviews were conducted. They were simply assessed pre-pandemic (this alone, as noted, is a strength).

2. Routines paragraph. The authors may be interested in this newly published article. Glynn, L. M., Davis, E. P., Luby, J. L., Baram, T. Z., & Sandman, C. A. (2021). A predictable home environment may protect child mental health during the COVID-19 pandemic. Neurobiology of Stress, 14, 100291.

3. Marginally significant findings. Given the number of contrasts run, (and once corrections are implemented these will be less significant), these should not be discussed.

4. Limitations. Self-report measures can be inaccurate – how? This won’t be obvious to all readers. Another limitation is the change in measurement from pre to during the pandemic (CBCL to SDQ). Also, the sample size, which cuts across two different developmental periods, lacks some power.

5. Given that the authors “pitch” in the intro that identifying simple and practical strategies that are easily accessible, inexpensive, and require no specialized resources outside the home” could be informative, I expected to see a practical implications section in the discussion. While some suggestions are provided throughout the discussion, a designated section for the implications for the simple and practical strategies recommended to mitigate subsequent risk is welcome.

6. PLOS authors have the option to publish the peer review history of their article (what does this mean?). If published, this will include your full peer review and any attached files.

Reviewer #1: No

Reviewer #2: No

---

## [Author Response · Author response to Decision Letter 0]

28 Jun 2021

Comments to the Author

** Denotes beginning of author response, *** denotes end of author response

1. Is the manuscript technically sound, and do the data support the conclusions?

Reviewer #1: Partly

Reviewer #2: Partly

2. Has the statistical analysis been performed appropriately and rigorously? 

Reviewer #1: Yes

Reviewer #2: No

3. Have the authors made all data underlying the findings in their manuscript fully available?

Reviewer #1: Yes

Reviewer #2: Yes

4. Is the manuscript presented in an intelligible fashion and written in standard English?

Reviewer #1: Yes

Reviewer #2: Yes

5. Review Comments to the Author

Reviewer #1: The present manuscript investigated longitudinal effects of stressors related to the COVID-19 pandemic on youth’s (7-15 years) psychopathological development as well as possible protective factors. Stressors were positively related to internalizing and externalizing symptoms half a year later. Exercise, a structured routing, and less screen time were related to less psychopathological symptoms. Authors discuss practical steps promoting the positive adjustment of families and youth to the stressful environment of the COVID-19 pandemic.

Overall evaluation

The manuscript addresses an important practical question on the psychological factors benefitting youth during the exposure to COVID-19 related stressors. However, the manuscript suffers from limitations mainly concerning theoretical background, hypotheses, methodology, and the results.

Major Concerns

Literature Review: The review presents a host of findings linking stress factors to youth’s psychopathological symptoms. However, the deduction of hypotheses seems largely theoretically unmotivated. Given the uniquely stressful environment caused by the COVID-19 pandemic, developmental theories should be included to predict expected effects and age differences. For example, while the inclusion of potentially protective factors understandably relies on practicability, it would be important to get more insights into their relation to developmental theories.

**

We appreciate the reviewer highlighting the need for a stronger theoretical framing of this manuscript. We have edited both the introduction and the discussion to reflect theoretical frameworks of resilience across development (Masten et al., 2021). In particular, we discuss how resilience factors exist across different levels from the individual, family and home, to the school and neighborhood. Importantly, during the early period of the COVID-19 pandemic when our first assessments were acquired schools were closed, stay at home orders were in place, and many community resources were also shuttered. As a result, children were cut off from many common sources of resilience, particularly those occurring at the school and neighborhood levels, as well as those stemming from interactions with peers. Therefore, we focus in this paper on home and family level sources of resilience. We chose specific factors that have previously been shown to be associated with better mental health outcomes or to buffer against the effect of stress on youth psychopathology. Additionally, we focused on specific behaviors that could still be engaged in even during the restrictive period of early stay-at-home orders of the pandemic. We have updated the introduction to reflect this framework. 

Page 4, Line 80 – Page 5, Line 103 :

“Leading theoretical models of resilience posit that factors that promote resilience exist across multiple levels including the individual, family, school, community, and broader cultural systems (24–26). Critically, during the early period of the COVID-19 pandemic when schools were closed, stay at home orders were in place, and many community resources were shuttered, children were cut off from many common sources of resilience, particularly those occurring at the school and community levels. As such, home and family-level factors may have been of even greater importance than in normal circumstances. Furthermore, given the constraints faced by many families with children, we focus on a set of simple and practical strategies that are easily accessible, inexpensive, and require no specialized resources or services outside the home. We selected factors that have previously been associated with reduced child psychopathology or buffer against mental health problems following exposure to stressors, including: higher levels of physical activity(27–29); access to nature and the outdoors(30–33); a consistent daily routine providing structure and predictability(16,34); getting a sufficient amount of sleep, which is often disrupted following stressors(35–37); and lower levels of passive screen time and news media consumption, given that higher use has been associated with elevations in child psychopathology(38), particularly following community-level stressors, like terrorist attacks(39–42).We also assessed the degree to which youth engaged in adaptive coping strategies during times of distress (e.g., exercising, seeking support from loved ones, or practicing mindfulness or meditation)(43–45). Finally, providing help for others in need is associated with reduced anxiety and depression(46,47). Here, we evaluated whether these nine simple and inexpensive strategies are (a) associated with reduced psychopathology symptoms during the pandemic and (b) buffer against the negative mental health consequences of pandemic-related stressors in children and adolescents.” 

***

In addition, hypotheses should be more clearly specified with respect to the expected effects, especially concerning potential age differences between children and youth.

**

Given the unique context of the pandemic and lack of prior data on similar long-term disruptions in children’s daily lives, we didn’t have strong expectations of which of these specific factors would be particularly beneficial to children or adolescents. However, we did hypothesize that adolescents would show a stronger association between pandemic-related stress and psychopathology given previous work that shows that adolescents is a period of particular vulnerability to mental health problems following stressful life events (Grant, 2003, 2004, Espejo 2007, Larson & Ham 1993, Monroe, 1999). We highlight this in the updated version of the manuscript. 

Page 6 Lines 124 - 129:

“Given the unique context of the COVID-19 pandemic, we did not have strong hypotheses about which particular protective factors would be more beneficial to children or adolescents. However, we did hypothesize that adolescents would show a stronger association between pandemic-related stress and psychopathology given previous work that shows that adolescence is a period of particular vulnerability to mental health problems following stressful life events (6,7,48–50).”

***

Finally, it seems questionable to call the follow-up measurement a post-pandemic assessment, given that many stressors may persist into this timeframe. Authors should give a more detailed overview over the respective pandemic-related restrictions and social-distancing measures so that one can get an idea of the exposure to stressors during the second measurement point. Otherwise the findings might be explained by youth’s increasing adaptation to the ongoing pandemic as much as by a change in pandemic-related stressors.

**

We agree with the reviewer that the use of the term “post-pandemic” implied that the pandemic was not still ongoing. This is inaccurate for both T1 (April – May 2020) and T2 (November 2020 – January 2021). During T2, while the strict stay-at-home orders had been lifted in the Seattle area, cases had spiked again, mask mandates and social distancing measures were still in place, and schools were still operating virtually. 

We have thus edited the title accordingly: “Promoting youth mental health during COVID-19: A Longitudinal Study” 

Additionally, we provide a supplemental table (Supplemental Table 1) with additional information about pandemic-related stressors (by category) at T2. Importantly we only asked about new stressors at T2 (since the last time we had contacted them). So, if for example, a participant had family member who became ill with COVID-19 in April 2020, this would be counted in the pandemic-related stressors at T1, but not at T2. 

We have updated the manuscript accordingly. 

Methods: 

Page 5, Lines 114-116:

“During this second follow-up, schools in the Seattle area were still operating virtually, social distancing guidelines were still in place, and new COVID-19 cases had reached a second peak.”

Page 9, Lines 205 – 210:

“We also assessed pandemic-related stressors at T2. Importantly we only asked about stressors occurring between T1 and T2. If, for example, a participant had family member who became ill with COVID-19 in April 2020, this would be counted in the pandemic-related stressors at T1, but not at T2. We used pandemic-related stressors at T1 in all analyses (including prospective analyses) but report on pandemic-related stressors at T2 to illustrate the ongoing nature of the pandemic during the second wave of data collection.” 

***

Method: The creation of the scales assessing potential protective factors and pandemic-related stressors should be described in more detail (e.g., reliability). 

**

Given that the COVID-19 pandemic presented a wide range of unique stressors that have not occurred in prior community-wide disruptions (e.g., natural disasters, terrorist attacks), it was necessary to create a novel measure to assess these types of experiences. It is standard practice in the field to do so when novel events occur for which existing stress measures do not adequately capture the full extent of specific types of stressful experiences. For example, this approach was taken by many research groups to study the impact of hurricane-related stressors after Hurricane Katrina (Galea et al., 2007; Mclaughlin et al., 2009; Galea et al., 2002; Pfefferbaum et al., 2000). Although existing measures of hurricane-related stressors existed, many failed to capture common experiences that were unique to Hurricane Katrina (e.g., having to sleep in the Superdome), and thus novel measures tailored to that particular event were developed. The same approach was required here, given the absence of appropriate existing measures on exposure to pandemic-related stressors, such as acquiring COVID-19, having a family member develop severe COVID-19 or die from the disease, prolonged exposure to remote schooling, etc.

We state this in the methods section (Page 8, Lines 197 - 184): 

“Given that the COVID-19 pandemic presented a wide range of unique stressors that have not occurred in prior community-wide disruptions, it was necessary to create a novel measure to assess these types of experiences. It is standard practice in the field to do so when novel events occur for which existing stress measures do not adequately capture the full extent of specific types of stressful experiences (e.g., to understand the unique hurricane-related stressors that occurred during Hurricane Katrina or experiences specific to the terrorist attacks on September 11th or the Oklahoma City bombing(12,41,54,55).

As a measure assessing the occurrence of distinct stressful life experiences, internal consistency between items is not an appropriate measure of reliability or validity, as the occurrence of one such stressor does not inherently relate to the probability of experiencing other distinct events. Indeed, internal consistency is not a metric typically reported for these types of stressful life events scales ( e.g., see Evans et al., 2013). However, the associations presented between the number of stressors and internalizing problems provide strong evidence of convergent validity. In addition, we find that the number of stressors is moderately associated with a measure of perceived stress as measured by the Perceived Stress Scale in this sample (r=0.399). This value is similar to the correlation between stressful life events and perceived stress observed in the original validity studies used to create the perceived stress scale (Cohen et al., 1983) (r=0.24-.35 among college students), providing further evidence of convergent validity of our pandemic-related stress measure. We also include this in the methods section (Page 9, Lines 200-204): 

“Here, we provide additional evidence for convergent validity by showing that the number of stressors is moderately associated with a measure of perceived stress as measured by the Perceived Stress Scale in this sample (r=0.399). This value is similar to the correlation between stressful life events and perceived stress observed in the original validity studies used to create the Perceived Stress Scale (r=0.24-.35)(57).”

***

Given that protective factors seem less pandemic-specific than the stressors, why did authors not rely on an established scale?

**

The core motivation in asking about protective factors within the COVID Experiences survey rather than relying on established scales was participant burden. While validated scales do exist for many of the protective factors (e.g. Pittsburgh Sleep Quality Index, Physical Activity Questionnaire for Children, Media Parenting Practices, Family Routines Inventory, German Coping Questionnaire for Children and Adolescents, etc), each of these surveys is at least 20 questions in length, and often longer. If we assume that a child can complete 5 questions per minute (which is a fairly liberal estimate for the younger children included in our sample), adding detailed measures for each of 9 potentially protective factors would have added at least 40 additional minutes to the survey, and probably more for many children. Obtaining survey data from children and families during the early phase of the pandemic was exceedingly challenging given the enormous additional burdens many families were facing along with a lack of access to childcare. Many parents expressed interest in completing the surveys but articulated concern about whether they could find the time to do so, even with the relatively brief nature of our assessments of many domains. Adding in additional items to the survey would have significantly increased the burden on both parents and children when families were navigating the very stressful and unpredictable period at the beginning of the pandemic and would have surely reduced the response and completion rate. We made the decision to limit the length of all of our surveys to increase acceptability for families and maximize our sample size. 

As we described above in response to your question about our measure of pandemic-related stressors, it is common practice to deploy novel measures to assess stressors, protective factors, and other salient environmental experiences during disasters, terrorist attacks, and other types of community-level disruptions (Galea et al., 2002; Galea et al., 2007; Mclaughlin et al., 2009, Pfefferbaum et al., 2000). This practice reflects the inherent trade-offs of using previously established measures that have been validated but do not capture the unique circumstances of novel events that may be important in shaping mental health along with participant burden during distressing situations where participating in research is not necessarily a priority.

However, we agree with the reviewer that this is a limitation of our study and thus have added this to our limitations section. 

Page 21 Lines 498-504: 

“Relatedly, we asked questions about potential protective factors in our COVID Experiences Survey, rather than using longer validated scales for each of the factors (e.g. Pittsburgh Sleep Quality Index, Physical Activity Questionnaire for Children, Media Parenting Practices, Family Routines Inventory, German Coping Questionnaire for `Children and Adolescents, etc.). This choice was made to maximize the information gained about each family, while minimizing participant burden and thus maximizing our sample size.”

***

What was the rationale behind selecting the highest values of caregiver or child on the CBCL and the SDQ? This seems at odds with the way theses scales are usually used and introduces the problem that some values are self-reported (if youth values were chosen) while other values are reports from caregivers.

**

The use of the higher parent or child report on the CBCL/YSR or SDQ is an implementation of the standard “or” rule used in combining parent and child reports of psychopathology. In this approach, if either a parent or child endorses a particular symptom it is counted with the assumption that if a symptom is reported, it is likely present. Thus, the reporter endorsing the higher level of symptoms or impairments is used. This is a standard approach in the literature on child psychopathology – for example it is how mental disorders are diagnosed in population-based studies of psychopathology in children and adolescents (e.g. Kessler et al., 2012; Merikangas et al., 2010). It also reflects the fact that parents are more valid reporters on some symptom domains—particularly externalizing behaviors like oppositionality and rule-breaking for which children typically under-report symptoms, whereas children are more valid reporters on other domains—particularly internalizing symptoms like worries, fears, and sad mood for which parents may be unaware and often under-report (e.g., Cantwell et al., 1997).

We have added further justification of the use of this method to the text. 

Methods: Page 10 Lines 232 - 238

“The use of the higher parent or child report for psychopathology is an implementation of the standard “or” rule used in combining parent and child reports of psychopathology. In this approach, if either a parent or child endorses a particular symptom it is counted with the assumption that if a symptom is reported, it is likely present. This is a standard approach in the literature on child psychopathology – for example it is how mental disorders are diagnosed in population-based studies of psychopathology in children and adolescents(60,61). ”

***

In addition, why was the measure of psychopathology not kept constant from the pre-pandemic assessment? The comparison with pre-pandemic levels of psychopathology would be more informative if both measurement points had used the same measure.

**

While we agree with the reviewer that it would have been ideal to use the same measure for psychopathology across all time points, again this decision came down to minimizing participant burden. The CBCL/YSR is 113 items while the SDQ is 25 items. Despite being shorter, the SDQ is highly reliable, has been validated against the CBCL/YSR, and is widely used in the U.S. and globally (Goodman et al., 1998, 1999; Klasen et al., 2000; Van Roy et al., 2008). As stated above, we were cognizant of minimizing burden on both parents and children when families were navigating the very stressful and unpredictable period at the beginning of the pandemic that involved a loss of access to school, childcare, and routine that placed enormous burden on families. Therefore, we decided to use another well-validated measure that strongly correlates with the CBCL/YSR (see Goodman et al., 1999; Klasen et al., 2000). Furthermore, we demonstrate moderate to high correlations between baseline internalizing and externalizing psychopathology (measured with the CBCL / YSR) and internalizing and externalizing psychopathology (measured with the SDQ) during the pandemic (r = .281 and r = .321, respectively, see Supplemental Table 3).

However, we agree with the reviewer that ideally, we would have used the same measures of psychopathology during the pandemic as prior to the pandemic. Thus, we have added this as a limitation. 

Page 21, Line 544-549

“Third, we used a different measure of psychopathology prior to the pandemic (CBCL/YSR) than after the onset of the pandemic (SDQ). While it would have been ideal to have the same measure at all time points, the CBCL/YSR is much longer than the SDQ and we were focused on minimizing participant burden during a period of time when families were facing numerous stressors and loss of access to typical childcare options. Thus, we chose to use a shorter questionnaire that is strongly correlated with the CBCL/YSR(62,65,79,80).”

***

Results: Running separate analyses for each of the protective factors seems questionable and also gives rise to the problem of inflating the alpha-error probability. Perhaps one could combine protective factors based on their correlational structure to obtain less factors overall. This seems also more feasible with the composite score of pandemic-related stressors that authors describe in the methods.

**

While we agree with the reviewer that the use of separate analyses for each protective factor may not be ideal given the number of factors of interest, the approach the reviewer suggests would result in loss of specificity regarding which specific factors may be most helpful for families to implement. If for example we combined all nine factors together and found higher levels of the protective factors overall were associated with better mental health or buffered against the association between stress and psychopathology, it would be difficult to know which behaviors were driving these differences. This would make it difficult for us to make suggestions for behaviors for families to implement. 

Moreover, the majority of these protective factors were not highly correlated with one another (absolute rs = .001 - .181) with the exception of physical activity & nature (.364), physical activity & outdoor time (r = .392), physical activity & coping (r = .247), screen time and sleep (r = -.252) screen time & news (r = .55), and news & coping (r = .203). However, the fact that the vast majority of these associations were very weak indicates that a latent factor would not adequately represent the constructs of interest.

However, in the revision of the manuscript we employ correction for multiple comparisons at the level of hypothesis to address the issue of multiple comparisons.

***

In addition, it would be interesting to see how levels of psychopathology changed from pre-pandemic to T1 to get insights into a possible increase in psychopathological symptoms after the onset of the pandemic.

**

Thank you for highlighting this important point. Prior to the pandemic 71 participants (31.7%) of the sample met subclinical criteria for internalizing problems and 39 participants (17.4%) met subclinical criteria for externalizing problems. After the onset of the pandemic there was a stark increase in both internalizing and externalizing psychopathology. Specifically, at T1, 127 (56.7%) met subclinical criteria for internalizing problems and 126 (56.2%) met subclinical criteria for externalizing problems. We have added this information to the results section to highlight the increase in symptoms after the onset of the pandemic. 

Page 12, Lines 277-283: 

“Prior to the pandemic, 71 participants (31.7% of the sample) were in the subclinical or clinical range for internalizing problems and 39 participants (17.4% of the sample) were in the subclinical or clinical range for externalizing problems. Internalizing and externalizing symptoms increased substantially during the early phase of the pandemic. Specifically, 127 (56.7%) were in the subclinical or clinical range for internalizing problems and 126 (56.2%) were in the subclinical or clinical range for externalizing problems at the beginning of the pandemic.”

***

Minor

p. 6 line 135: typo - drawn

p. 6 lines 130 and 139: “their” instead of “a” careviger

**

Thank you for these corrections. We have edited them in the manuscript. 

***

Reviewer #2: Thank you for the opportunity to review this interesting, timely, and well-written manuscript. In this paper, the authors provide a thoughtful examination of which protective (and easily modifiable) factors in the child’s ecology may mitigate internalizing and externalizing symptom distress during the COVID-19 pandemic. The introduction is well formulated and provides a good overview of the literature. There are many strengths to this paper, including its longitudinal design with pre-post measurements and two COVID-19 time points, practical approach to identifying modifiable protective factors, and multi-informant data. However, my enthusiasm was tempered by the amalgamation of two samples varying by developmental period and some of the analytic choices. Below I provide some suggestions to increase clarity, transparency, and impact.

Certainly, a major strength of the paper is the availability of pre-pandemic internalizing and externalizing symptoms and controlling for these in analyses. While the authors provide strong justification in the introduction for why stress should increase during COVID-19 and for their moderator variables, I do think it should be noted in the intro that the strongest predictor of pandemic internalizing/externalizing problems is likely to be pre-pandemic mental health difficulties. This is implied but not explicitly stated in the introduction and is evident when exploring the correlation matrix where associations between baseline (pre-pandemic) and COVID-19 internalizing and externalizing are strong (and much larger than any of the other variables included). This finding should also be noted at the outset of the results and/or in the discussion because it contextualizes the findings.

**

Thank you for pointing this out. We agree that this is a crucial point and now highlight it more explicitly in the introduction. 

Page 5, Lines 105-110

“We examined these questions by combining two longitudinal samples of children and adolescents whose mental health was assessed prior to the COVID-19 pandemic in Seattle, Washington. This aspect of this study is critical because one of the strongest predictors of psychopathology during the pandemic is likely to be psychopathology prior to the pandemic. By controlling for pre-pandemic psychopathology, we are able to investigate changes in psychopathology that occurred during the pandemic.”

***

The authors aptly note several factors that may be associated with child stress and increased behavior problems during COVID-19. These could all independently predict child outcomes. What is the justification for combining these into a cumulative risk composite? From a developmental psychopathology perspective, I understand, but the justification is needed nonetheless, especially for those less familiar with cumulative risk.

**

Thank you for this comment. We agree that additional explanation is helpful. We have added the following text to the introduction (Page 8, Line 207 – Page 9, Line 209):

“Importantly, many previous studies demonstrate the utility and convergent validity of cumulative stress measures in relation to health outcomes, with a greater number of stressors predicting higher levels of mental and physical health problems (56).”

However, there are of course limitations of cumulative stress measures and we have added this to the section of the paper on Page 22, Lines 544 – 549:

“Fifth, we demonstrate the predictive validity of the pandemic-related stress measure via moderate associations with psychopathology at both waves as well as a measure of perceived stress. However, this cumulative risk approach is limited in that it weights stressors equally that could have variable impacts. Future work should investigate whether specific stressors have been more strongly linked to changes in mental health during the pandemic (see Supplemental Table 4 for associations of specific stressors and psychopathology at T1 and T2).”

***

It is not until the methods that one discovered the sample is in fact two separate longitudinal samples. Greater justification is needed to combine two different samples, especially with such different age ranges and developmental periods. The authors should be more transparent in the abstract and the introduction about this fact, and also should justify why these samples should be combined (other than to increase sample size).

**

Thank you for this critique. We now make it clearer that the sample used in the present study came from two separate ongoing studies. We justify the combination of these two samples for several reasons. First, this allowed us to explore how the pandemic impacted youth mental health across a wider age range. Second, while these youths came from different study samples, they were recruited using similar methods from the same general population; specifically, both studies recruited youths from the Seattle area using community-based methods aimed at identifying a sample with a wide range of socioeconomic backgrounds. Additionally, we had the identical measures of pre-pandemic psychopathology on both samples. Finally, as the reviewer notes, combining the two samples also allowed us to increase our sample size. Additionally, as we detail below, the samples did not differ with regard to demographics, socioeconomic status, or exposure to pandemic-related stressors. The only meaningful difference in the samples was the age range.

We agree with the reviewer that it would have been ideal if we had had one larger sample that had spanned these ages. In the revised manuscript, we make it clearer in the abstract and introduction that we combined these data from two samples. And we now include this as a limitation in the discussion section. 

Abstract:

“We assessed pandemic-related stressors, internalizing and externalizing psychopathology, and potential protective factors by combining two longitudinal samples of children and adolescents (N=224, 7-10 and 13-15 years) assessed prior to the pandemic, during the stay-at-home orders, and six months later.”

Introduction (Page 5, Lines 105 – 107):

“We examined these questions by combining two longitudinal sample of children and adolescents whose mental health was assessed prior to the COVID-19 pandemic in Seattle, Washington.”

Limitations (Page 21, Line 528 – Page 22, Line 533): 

“Fourth, we combined data from two separate samples of children (aged 7-10 and 13-15 at T1). Both samples were recruited using similar methods from the same target population, and we had identical measures of pre-pandemic psychopathology on both samples. Moreover, the samples did not differ in demographics, SES, or exposure to pandemic-related stressors. However, using two samples with a gap in age limited our ability to understand age effects across the entire spectrum of childhood and adolescence.”

***

Methods

1. “174 of these youth” add (77.7% of original sample) so that attrition is clearly stated.

**

We have added this to the methods. We also realized there was a slight error in our calculation such that 184 of the original sample participated in the second wave, but 9 of those participants did not complete the mental health assessment. We have clarified this in the text. 

Page 6, Lines 148 – 151: 

“Six months later, 184 of these youth (82% of the initial pandemic sample) and a caregiver again completed an assessment of internalizing and externalizing symptoms. Ten participants did not complete these mental health assessments and therefore were excluded from analyses at T2.”

***

2. Do these two samples differ on any demographic or levels of the stress or protective factors? These should be tested and noted and sign differences should be included as covariates.

**

There were no significant differences between the samples with regards to income-to-needs or sex (ps > .8). Additionally, the two samples did not differ in their level of pandemic related stressor (p = .907). 

We now note this in the Methods section when describing the sample (Page 7, Line 162-165):

“These two samples came from the same general population (youth in the Seattle area from a wide range of socioeconomic backgrounds). Critically, these two samples did not differ with regards to socioeconomic status, as measured by the income-to-needs ratio, or sex (ps > .8) or in exposure to pandemic related stressors (p = .907).”

Moreover, the two samples did not differ on degree of engagement in many of the protective factors including physical exercise, structured routine, helping behaviors, or engaging in adaptive coping strategies (ps > .3). They did differ on some potential protective factors, in ways that would be expected given the different age range of the samples. For example, younger children spent significantly more nature and time outdoors than adolescents (ps < 0.03), while adolescents spent more time on screens and were more likely to consume more than 2 hours of news per day, and not get the recommended amount of sleep (ps < .001). 

Notably, this can be seen by the significant associations between age and these potential protective factors in the bivariate correlation matrix we report in the paper (see Supplemental Table 3). In the manuscript we control for age in all analyses and explore age-related differences in the associations between these factors and psychopathology. Therefore, we believe we have adequately addressed this issue. However, we do note the limitation of using these two samples rather than one larger sample that spanned the full age spectrum of interest in the limitations section. 

Limitations (Page 21, Line 528 – Page 22, Line 533) : 

“Fourth, we combined data from two separate samples of children (aged 7-10 and 13-15 at T1). Both samples were recruited using similar methods from the same target population, and we had identical measures of pre-pandemic psychopathology on both samples. Moreover, the samples did not differ in demographics, SES, or exposure to pandemic-related stressors. However, using two samples with a gap in age limited our ability to understand age effects across the entire spectrum of childhood and adolescence.

***

3. Going along with point #2, did all children provide self-reports of behavior problems? I am assuming that the 6-8 year olds did not pre-pandemic. This needs to be made clear as the multi-informant data is a strength, but it’s not clear how extensive this multi-informant data stretches.

**

Thank you for catching this. Indeed the 6-8 year olds did not complete the YSR. We have clarified that in the methods section. 

Page 10, Lines 244 – 246: 

“The children who were 6-8 years old at the pre-pandemic time point did not complete the YSR; only the CBCL was used to compute their pre-pandemic symptoms at that time point.” 

***

4. Why the switch from the CBCL to the SDQ?

**

While we agree with the reviewer that it would have been ideal to use the same measure for psychopathology across all time points, again this decision came down to minimizing participant burden. The CBCL/YSR is 113 items while the SDQ is 25 items. Despite being shorter, the SDQ is highly reliable, has been validated against the CBCL/YSR, and is widely used in the U.S. and globally (Goodman et al., 1998, 1999; Klasen et al., 2000; Van Roy et al., 2008). As stated above, we were cognizant of minimizing burden on both parents and children when families were navigating the very stressful and unpredictable period at the beginning of the pandemic that involved a loss of access to school, childcare, and routine that placed enormous burden on families. Therefore, we decided to use another well-validated measure that strongly correlates with the CBCL/YSR (see Goodman et al., 1999; Klasen et al., 2000). Furthermore, we demonstrate moderate to high correlations between baseline internalizing and externalizing psychopathology (measured with the CBCL / YSR) and internalizing and externalizing psychopathology (measured with the SDQ) during the pandemic(r = .281 and r = .321, respectively, see Supplemental Table 3).

However, we agree with the reviewer that ideally, we would have used the same measures of psychopathology during the pandemic as prior to the pandemic. Thus, we have added this as a limitation. 

Page 21, Lines 507 – 512: 

“Third, we used a different measure of psychopathology prior to the pandemic (Child Behavior Checklist/Youth Self Report) as we did after the onset of the pandemic (Strengths and Difficulties Questionnaire). While it would have been ideal to have the same measure at all time points, the CBCL is much longer than the SDQ and we were focused on minimizing participant burden during a period of time when families were facing numerous stressors and loss of access to typical childcare options. Thus, we chose to use a shorter questionnaire that is strongly correlated with the CBCL/YSR(59,74–76).”

***

5. Using the highest t-score for either rater – is there support for this in the literature? Why not average informants?

**

Both reviewers bring up this point which highlights the need for us to justify this choice in the manuscript. 

The use of the higher parent or child report on the CBCL/YSR or SDQ is an implementation of the standard “or” rule used in combining parent and child reports of psychopathology. In this approach, if either a parent or child endorses a particular symptom it is counted with the assumption that if a symptom is reported, it is likely present. Thus, the reporter endorsing the higher level of symptoms or impairments is used. This is a standard approach in the literature on child psychopathology – for example it is how mental disorders are diagnosed in population-based studies of psychopathology in children and adolescents (e.g. Kessler et al., 2012; Merikangas et al., 2010). It also reflects the fact that parents are more valid reporters on some symptom domains—particularly externalizing behaviors like oppositionality and rule-breaking for which children typically under-report symptoms, whereas children are more valid reporters on other domains—particularly internalizing symptoms like worries, fears, and sad mood for which parents may be unaware and often under-report (e.g., Cantwell et al., 1997). 

We have added further justification of the use of this method to the text. 

Methods: Page 10 Lines 246 – 252: 

“The use of the higher caregiver or child report for psychopathology is an implementation of the standard “or” rule used in combining caregiver and child reports of psychopathology. In this approach, if either a caregiver or child endorses a particular symptom it is counted with the assumption that if a symptom is reported, it is likely present. This is a standard approach in the literature on child psychopathology – for example it is how mental disorders are diagnosed in population-based studies of psychopathology in children and adolescents (60,61).”

***

6. What are the alphas for the behavioral problem measures?

**

Chronbach’s alpha demonstrated acceptable reliability for parent and child report of the conduct problems subscale of the SDQ (Chronbach’s alpha = 0.752 at T1 and alpha = 0.734 at T2).

***

7. With close to 25% of the sample lost at the COVID follow up, how was missing data handled?

**

Missing data was handled using listwise deletion for analyses at T2 for participants who did now complete the follow-up. We have added this to the methods.

Page 11, Lines 287 - 288:

“Listwise deletion was used to handle missing data at T2, excluding participants from analysis who did not complete the second follow-up during the pandemic.”

***

8. Adding child sex and pre-pandemic mental health is a strength of this paper. But why not add child ethnicity and family income, for of which predict the outcomes in this paper. I also think age should be added as a covariate (versus stratifying by age).

**

Age was included as a continuous covariate in all analyses in the original manuscript. We apologize for the oversight of not including this clearly in the manuscript and have now edited the paper to reflect this.

Furthermore, in the current version of the manuscript, we have added the income-to-needs ratio as a covariate in all analyses. We describe this in the methods and results section as follows: 

Methods (Page 10, Lines 259 – 256):

“Family Income: At T1, we asked caregivers to report their total combined family income for the 12 months prior to the onset of the pandemic in 14 bins. The median of the income bins was used except for the lowest and highest bins which were assigned $14,570 and $150,000, respectively. We then calculated the income-to-needs ratio by dividing the family’s income by the federal poverty line for a family of that size in 2020, with values less than one indicating income below the poverty line. Nine caregivers did not provide information on family income and were thus excluded from analyses. Median income-to-needs ratio was 4.19 (min = 0.35, max = 8.41).”

Page 11, Lines 281 – 283: 

“Continuous age, sex, income-to-needs ratio, and pre-pandemic symptoms measured using the CBCL/YSR prior to the pandemic were included as covariates in all analyses.”

***

9. Given the number of analyses run, the authors should adjust the p-value for multiple contrasts.

**

We have included false discovery rate (FDR) correction for multiple comparison at the level of hypothesis, controlling for each association at T1 and T2. We have added this to the methods section. 

Page 11, Lines 295 - 298:

“False discovery rate (FDR) correction was applied at the level of hypothesis such that we corrected for comparisons at T1 and T2 (e.g. association between physical activity and internalizing psychopathology at T1 and T2).”

***

Results

1. Are patterns of associations similar across the two samples?

**

We did not separately look at the associations between the two samples, but rather used age as a continuous moderator variable. Because the two groups had a gap in age between them (8-10 and 13-15), this a moderator analysis with age as a variable can also acts as an investigation of whether the associations varied by group. Including this as a moderator variable rather than running separate analyses is a more direct test of whether there is variability across groups and results in greater power to examine the associations of interest. As discussed in the results there were some significant age x protective factor and age x stress x protective factor results. 

While these youths came from different study samples, they were recruited using similar methods from the same general population; specifically, both studies recruited youths from the Seattle area using community-based methods aimed at identifying a sample with a wide range of socioeconomic backgrounds. Additionally, we had the identical measures of pre-pandemic psychopathology on both samples. Finally, as the reviewer notes, combining the two samples also allowed us to increase our sample size. Additionally, as we detail below, the samples did not differ with regard to demographics, socioeconomic status, or exposure to pandemic-related stressors. The only meaningful difference in the samples was the age range.

We note this in the description of the sample (Page 7, Lines 174 - 177:

“These two samples came from the same general population (youth in the Seattle area from a wide range of socioeconomic backgrounds). Critically, these two samples did not differ with regards to socioeconomic status, as measured by the income-to-needs ratio, or sex (ps > .8) or in exposure to pandemic related stressors (p = .907).”

We have also noted the limitation of combining two samples in the discussion: 

Page 22, Lines 533 - 538: 

“Fourth, we combined data from two separate samples of children (aged 7-10 and 13-15 at T1). Both samples were recruited using similar methods from the same target population, and we had identical measures of pre-pandemic psychopathology on both samples. Moreover, the samples did not differ in demographics, SES, or exposure to pandemic-related stressors. However, using two samples with a gap in age limited our ability to understand age effects across the entire spectrum of childhood and adolescence.”

***

2. “The number of pandemic-related stressors was strongly associated with increases in both internalizing (�=0.322 [0.211, 0.432], and externalizing symptoms (�=0.225 [0.136, 0.314], 208 p<.001), symptoms during the pandemic, controlling for pre-pandemic symptoms”. Can you add the estimates here for the pre-pandemic symptoms so that the magnitude of effects can be directly evaluated (by the reader). As far as I can tell, this data is not in any of the tables/figures. Please report the pandemic-related stressors to psychopathology analyses in a table and include covariates to ensure the findings hold with these variables included. This is especially important given the sample differences.

**

Thank you for highlighting the importance of including this information. We have added this to the beginning of the results section. 

Results Page 12, Line 323 – Page 13, Line 333: 

“The number of pandemic-related stressors was strongly associated with increases in both internalizing (�=0.345, p < .001), and externalizing symptoms (�=0.297, p<.001) symptoms during the pandemic, controlling for pre-pandemic symptoms. As expected, pre-pandemic symptoms were also strongly associated psychopathology during the pandemic in this model (�=0.279, p < .001 and �= 0.296, p < .001 for internalizing and externalizing psychopathology, respectively). 

Similarly, the number of pandemic-related stressors early in the pandemic was positively associated with internalizing (�=0.243, p = .001) and externalizing (�=0.288, p < .001) symptoms later in the pandemic, controlling for pre-pandemic symptoms. Again, pre-pandemic symptoms were strongly associated with internalizing and externalizing problems at T2 (�=0.260, p = .001 and �= 0.278, p < .001, respectively). ”

***

3. Are the stratification analyses by age simply a stratification by the two samples? If so, they should be presented as such for more clarity and age as a continuous measure should be used a co-variate in analyses.

**

We used age as a continuous covariate in all analyses. Additionally, when looking at age x protective factor interactions and age x stressor x protective factor interactions, we also used age as a continuous moderator variable. We did stratify by sample only for the simple slopes analyses and for visualization purposes of significant interactions. 

We more clearly and explicitly state this in the manuscript

Page 11, Line 281 – 283: 

“Continuous age, sex, income-to-needs ratio, and pre-pandemic symptoms measured using the CBCL/YSR prior to the pandemic were included as covariates in all analyses.”

Page 11 Lines 292 – 295:

“Stratification for simple slope analyses in analyses that used continuous moderators were conducted using a median split. In the case of age analyses, because there was a gap in age between the oldest children (10 years) and the youngest adolescents (13 years), stratifying by sample for these purposes was equivalent to stratifying by a median split.”

***

Discussion

1. “mental health had been carefully assessed prior to the pandemic” remove carefully here as that implies diagnostic interviews were conducted. They were simply assessed pre-pandemic (this alone, as noted, is a strength).

**

Thank you for this critique. We have removed “carefully” from that sentence. 

***

2. Routines paragraph. The authors may be interested in this newly published article. Glynn, L. M., Davis, E. P., Luby, J. L., Baram, T. Z., & Sandman, C. A. (2021). A predictable home environment may protect child mental health during the COVID-19 pandemic. Neurobiology of Stress, 14, 100291.

**

Thank you for calling our attention to this very relevant paper! We have added this citation to the discussion of the benefits of maintaining a structured routine:

Page 17, Line 446 – Page 18, Line 452:

“Moreover, a recent paper during the pandemic showed that preschoolers in families that maintained a structured routine during the pandemic showed lower rates of depression and externalizing problems, over and above the effect of food insecurity, socioeconomic status, dual-parent status, maternal depression, and stress (68). Our current findings extend this work by demonstrating that a structured routine may also be important for older children and adolescents.”

***

3. Marginally significant findings. Given the number of contrasts run, (and once corrections are implemented these will be less significant), these should not be discussed.

**

We appreciate the reviewer’s encouragement to remove marginally significant findings and have carefully considered their request. We note that in the revision, we control for income-to-needs and employ correction for multiple comparison at the level of hypothesis. This changes some of the findings from significant to marginal and some that were previously marginal to no longer significant. We have removed any discussion of effects that are no longer significant (e.g. association between physical exercise and psychopathology) and for the most part do not report or discuss marginal findings.

However, we would like to note that fully removing any discussion of marginal effect would result in a paper that only discusses screen time, news consumption, and having a structured daily routine. As we note in the discussion, the findings of the impact of screen time are mixed in the field. And we would like to avoid publishing a paper that focuses mostly on what families should not do (spend time on screens and consuming news) and also include some proactive steps that families can take (implement a structured routine, spend time in nature, get the recommended amount of sleep, etc.). As such, we have noted just two marginal findings, because they highlight strategies that could be beneficial for families (e.g., spending time in nature, getting the recommended amount of sleep) that do not simply involve avoiding things like screen time or news consumption.

Furthermore, we wish to highlight the difference between statistical significance and practical significance. A change in a p-value from statistically significant to marginally significant because of correction for multiple comparisons does not necessarily indicate that that factor is irrelevant to families. As such, we believe it would be a disservice to the field and the study’s findings to publish a paper that only discusses these results when the focus of the paper could make a much broader and more nuanced contribution.

Consequently, we have incorporated the spirit of the reviewer’s comment and have ensured that any discussion of marginally significant effects are phrased speculatively and are explicitly noted as marginal. We hope that these revisions provide readers with a transparent discussion of the paper’s findings, both significant and marginal, with the goal of providing a range of possible strategies families can employ to promote better mental health in their children during this stressful and challenging time. 

***

4. Limitations. Self-report measures can be inaccurate – how? This won’t be obvious to all readers. Another limitation is the change in measurement from pre to during the pandemic (CBCL to SDQ). Also, the sample size, which cuts across two different developmental periods, lacks some power.

**

We have added these limitations to the discussion. 

Page 21, Line 522 - 526: 

“The present study has several limitations which should be acknowledged. First, we relied on self-report measures of behavior, which can be inaccurate due to recall bias. Future studies may benefit from using actigraphy to assess physical activity and sleep, geolocation to measure time spent in nature and outdoors, and direct reports of screen time use and news media consumption from digital devices for more accurate measures of potential protective factors.”

Page 21, Lines 530 – 535:

“Third, we used a different measure of psychopathology prior to the pandemic (CBCL/YSR) than after the onset of the pandemic (SDQ). While it would have been ideal to have the same measure at all time points, the CBCL/YSR is much longer than the SDQ and we were focused on minimizing participant burden during a period of time when families were facing numerous stressors and loss of access to typical childcare options. Thus, we chose to use a shorter questionnaire that is strongly correlated with the CBCL/YSR(62,65,79,80).”

Page 22, Lines 542 – 547:

“Fourth, we combined data from two separate samples of children (aged 7-10 and 13-15 at T1). Both samples were recruited using similar methods from the same target population, and we had identical measures of pre-pandemic psychopathology on both samples. Moreover, the samples did not differ in demographics, SES, or exposure to pandemic-related stressors. However, using two samples with a gap in age limited our ability to understand age effects across the entire spectrum of childhood and adolescence.”

***

5. Given that the authors “pitch” in the intro that identifying simple and practical strategies that are easily accessible, inexpensive, and require no specialized resources outside the home” could be informative, I expected to see a practical implications section in the discussion. While some suggestions are provided throughout the discussion, a designated section for the implications for the simple and practical strategies recommended to mitigate subsequent risk is welcome.

**

We have updated the conclusions section of the paper to be “Conclusions and Practical Implications” that includes the main takeaways of the paper. 

Page 22 Lines 559 – 565: 

“Conclusions and Practical Implications

We identify practical and easily accessible strategies that may promote greater well-being for children and adolescents during the COVID-19 pandemic. Based on these findings, we suggest that parents encourage youth to develop a structured daily routine, limit passive screen time use, limit exposure to news media—particularly for young children, and to a lesser extent spend more time in nature, and encourage youth to get the recommended amount of sleep.”

***

6. PLOS authors have the option to publish the peer review history of their article (what does this mean?). If published, this will include your full peer review and any attached files.

Do you want your identity to be public for this peer review? For information about this choice, including consent withdrawal, please see our Privacy Policy.

Reviewer #1: No

Reviewer #2: No

---

## [Decision Letter · Decision Letter 1]

14 Jul 2021

Promoting youth mental health during COVID-19: A Longitudinal Study

PONE-D-21-09922R1

Dear Dr. Rosen,

We’re pleased to inform you that your manuscript has been judged scientifically suitable for publication and will be formally accepted for publication once it meets all outstanding technical requirements.

Kind regards,

Helena R. Slobodskaya, M.D., Ph.D., D.Sc.

Academic Editor

PLOS ONE

Additional Editor Comments (optional):

Reviewers' comments:

Reviewer's Responses to Questions

**Comments to the Author**

1. If the authors have adequately addressed your comments raised in a previous round of review and you feel that this manuscript is now acceptable for publication, you may indicate that here to bypass the “Comments to the Author” section, enter your conflict of interest statement in the “Confidential to Editor” section, and submit your "Accept" recommendation.

Reviewer #1: All comments have been addressed

Reviewer #2: All comments have been addressed

2. Is the manuscript technically sound, and do the data support the conclusions?

Reviewer #1: Yes

Reviewer #2: Yes

3. Has the statistical analysis been performed appropriately and rigorously? 

Reviewer #1: Yes

Reviewer #2: Yes

4. Have the authors made all data underlying the findings in their manuscript fully available?

Reviewer #1: Yes

Reviewer #2: Yes

5. Is the manuscript presented in an intelligible fashion and written in standard English?

Reviewer #1: Yes

Reviewer #2: Yes

6. Review Comments to the Author

Reviewer #1: (No Response)

Reviewer #2: The authors of done an excellent job responding to the reviewer comments and queries. The only remaining issue for me is the use of listwise deletion to account for missing data, which removes all cases with any missing data (due to attrition or failing to complete a questionnaire in full). There are critiques of this method, most notably that missing data may not be at random. Can the authors determine if the data is missing completely at random (MCAR)? If so, I suggest they use a different missing data approach (such as FIML).

7. PLOS authors have the option to publish the peer review history of their article (what does this mean?). If published, this will include your full peer review and any attached files.

Reviewer #1: No

Reviewer #2: No

---

## [Editor Report · Acceptance letter]

3 Aug 2021

PONE-D-21-09922R1 

Promoting youth mental health during the COVID-19 pandemic: A Longitudinal Study 

Dear Dr. Rosen:

I'm pleased to inform you that your manuscript has been deemed suitable for publication in PLOS ONE. Congratulations! Your manuscript is now with our production department. 

Kind regards, 

on behalf of

Dr. Helena R. Slobodskaya 

Academic Editor

PLOS ONE